# Spatiotemporal responses in crop water footprint and benchmark under different irrigation techniques to climate change scenarios in China

Zhiwei Yue[1, 3, *], Xiangxiang Ji[1, 3, *], La Zhuo[2, 3, 4, 5], Wei Wang[4,5], Zhibin Li[4,5], Pute Wu[2, 3, 4, 5]

[1]College of Water Resources and Architectural Engineering, Northwest A&F University, Yangling 712100, China
[2]Institute of Soil and Water Conservation, Northwest A&F University, Yangling 712100, China
[3]Institute of Water-saving Agriculture in Arid Regions of China, Northwest A&F University, Yangling 712100, China
[4]Institute of Soil and Water Conservation, Chinese Academy of Sciences and Ministry of Water Resources, Yangling 712100, China
[5] University of Chinese Academy of Sciences, Beijing 100049, China.

*The authors contribute equally.
*Correspondence to*: La Zhuo (zhuola@nwafu.edu.cn; lzhuo@ms.iswc.ac.cn) and Pute Wu (gjzwpt@hotmail.com)

**Abstract.** Adaptation to future climate change with limited water resources is a major global challenge to sustainable and sufficient crop production. However, the large-scale responses of crop water footprint and its associated benchmarks under various irrigation techniques to future climate change scenarios remain unclear. The present study quantified the responses of maize and wheat water footprint per unit yield (WF, $m^3 t^{-1}$) and corresponding WF benchmarks under two representative concentration pathways (RCPs) in the 2030s, 2050s, and 2080s at a 5-arc minute grid level in China. The AquaCrop model with the outputs of six global climate models in Coupled Model Intercomparison Project Phase 5 (CMIP5) as its input data was used to simulate the WF of maize and wheat. The differences among rain-fed and furrow-, micro-, and sprinkler-irrigated wheat and maize were identified. Compared with the baseline year (2013), maize WF will increase under both RCP2.6 and RCP8.5, by 17 % and 13 %, respectively, until the 2080s. Wheat WF will increase under RCP2.6 (by 12 % until the 2080s) and decrease by 12 % under RCP8.5 until the 2080s, with a higher increase in wheat yield and decrease in wheat WF due to the higher $CO_2$ concentration in 2080s under RCP8.5. WF will increase the most for rain-fed crops. Relative to rain-fed crops, micro irrigation and sprinkler irrigation result in the smallest increases in WF for maize and wheat, respectively. These water-saving managements will mitigate the negative impact of climate change more effectively. The WF benchmarks of maize and wheat in the humid zone (~overall average at 680 $m^3 t^{-1}$ for maize and 873 $m^3 t^{-1}$ for wheat at 20[th] percentile) are 13–32 % higher than those in the arid zone (~ overall average at 601 $m^3 t^{-1}$ for maize and 753 $m^3 t^{-1}$ for wheat). The differences in WF benchmarks among various irrigation techniques are more significant in the arid zone, which can be as high as 57%, for 20[th] percentile WF benchmarks of 1020 $m^3 t^{-1}$ for sprinkler-irrigated wheat and 648 $m^3 t^{-1}$ for micro-irrigated wheat. Nevertheless, WF benchmarks will not respond to climate changes as dramatically as the WF in the same area, especially in the area with limited agricultural development. The present study demonstrated that the visible different responses to climate change in terms of crop water consumption, water use efficiency, and WF benchmarks under different irrigation techniques cannot be

ignored. It also lays the foundation for future investigations into the influences of irrigation methods, RCPs, and crop types on
WF and its benchmarks in response to climate change in all agricultural regions worldwide.
**1 Introduction**
The progressive decline in water resource availability is a major impediment to global food production security (Pastor
et al., 2019; Trnka et al., 2019; Konapala et al., 2020). Food crops are the main source of human nutrition (Myers et al., 2017;
Lobell and Gourdji, 2012). Humans depend on food crops for ~47 % of their daily protein intake (FAO, 2021). However, as a
result of human activity, the climate system is changing and global warming is a significant characteristic of this process (IPCC,
2021). Since the 1980s, each successive decade has been warmer than any preceding one after 1850 (Kappelle, 2020). Climate
change affects water consumption and crop yield by altering precipitation, temperature, carbon dioxide ($CO_2$) concentration,
and other factors during crop growth (Hatfield and Dold, 2019). Crop adaptation to future climate change with limited water
resources has become a major challenge in sustainable crop production and supply worldwide.
The water footprint per unit crop (WF, $m^3$ $t^{-1}$) (Hoekstra, 2003) is the amount of water consumed by the crop per unit
yield during crop growth within a certain region. It includes blue WF (surface and groundwater), green WF (precipitation that
will not become runoff), and grey WF (freshwater that assimilates pollutants from human activities) (Hoekstra et al., 2011).
Blue and green WF are collectively known as consumptive WF, and grey WF is also called degradative WF (Hoekstra, 2013).
Unlike traditional crop water productivity and other agricultural water metrics, WF covers water consumption, sources, and
spatiotemporal dimensions during the crop growth period. Therefore, water consumption intensity and efficiency for irrigated
and rain-fed growing modes may be compared. WF is an effective indicator of the sustainability of regional water use and
optimal water resource allocation (Xu et al., 2019; Mali et al., 2021). The present study focuses exclusively on consumptive
WF, which depends on crop yield and the intensity of water consumption per unit planted area.
Several studies have been conducted on the responses of WF to future climate change. Nevertheless, no consensus has
been reached. Certain scholars believe that future climate change will weaken food crop production security. Ahmadi et al.
(2021) reported that maize WF in the Qazvin Plain of India will increase by 42 % and 147 % under representative concentration
pathways (RCP) 4.5 and RCP8.5, respectively, by 2061–2080. Zheng et al. (2020) found that rice yield in Henan and Jiangsu
Provinces (China) will decrease, while WF will increase under four RCPs at various stages of the 21[st] century. Other scholars
believe that crop yield may actually benefit from future increases in precipitation and atmospheric $CO_2$ concentration. Jans et
al. (2021) considered the combined effects of changes in climatic factors, such as temperature, precipitation, and rising
atmospheric $CO_2$ concentration, and predicted that between 2011 and 2099, global cotton yield will increase by > 50 % and
WF will decrease by 30 % under RCP8.5. Arunrat et al. (2020) found that in the present century, the yield of individual and
large-scale rice farms in Thailand will increase by 1–30 % and 2–31 %, respectively, while WF will decrease by 10–43 % and
1–67 %, respectively, under RCP4.5. Significant spatiotemporal differences in WF under various irrigation techniques have
been confirmed at the site (Chukalla et al., 2015) and regional (Wang et al., 2019) scales. However, current large-scale studies

on the responses of WF to environmental change are usually based on simulations assuming adequate furrow irrigation. These studies exclude comparisons between various irrigation techniques and the differences in their influences on crop WFs. Although Dai et al. (2020) optimised maize and wheat cropping patterns under RCP4.5 and RCP8.5 with consideration of various irrigation modes in the Huaihe River Basin in China by 2050, they only considered blue water.

Magnitudes and constitution of crop WF vary widely among regions and areas (Mekonnen and Hoekstra, 2011). To encourage water users to reduce WF to a reasonable level, Hoekstra (2013, 2014) recommended establishing WF benchmarks for different products as they facilitate prudent water allocation and fair water resource sharing among sectors and users (Hoekstra, 2013). On the large-scale, specific WF benchmarks can be set for crops grown on different farms within the same region (Mekonnen and Hoekstra, 2014). A previous study demonstrated the sensitivity of WF benchmarks to climate zones (Zhuo et al., 2016a). WF benchmarks significantly differ among irrigation techniques, especially in arid zones (Wang et al., 2019). However, little is known about the responses of WF benchmarks under different irrigation techniques to future climate change.

To investigate the influence of future climate change on large-scale WF and benchmarks under diverse irrigation techniques, maize and wheat grown in mainland China were the subjects of this study. We used the outputs of six global climate models (GCMs) (Table 1), including three models each for relatively wet and dry climate outputs, in Coupled Model Intercomparison Project Phase 5 (CMIP5). We then used the AquaCrop model to simulate the spatiotemporal responses of blue and green WF and corresponding WF benchmarks for wheat and maize in the 2030s (2020–2049), 2050s (2040–2069), and 2080s (2070–2099) under RCP2.6 and RCP8.5 at a 5-arc minute grid resolution. We distinguished between rain-fed and irrigated growing modes and among furrow, micro, and sprinkler irrigation.

As of 2019, China was the world's second largest maize and largest wheat producer, accounting for 23 % and 17 % of total global production, respectively (FAO, 2021). China's cereal production has helped stabilise global food production and supply. In 2019, the planted areas of maize and wheat in China were 41 million ha and 24 million ha, respectively, and accounted for 25 % and 14 % of the national total croplands, respectively (NBSC, 2021). Cereal production consumes substantial volumes of water in China, and these quantities change over time. Zhuo et al. (2019) reported that maize water consumption increased by 49 % between 2000 and 2013 as planted areas and feed demand increased. Conversely, Wang et al. (2019) reported that wheat planted and irrigated areas decreased and water consumption slightly declined (4.4 %) from 2000 to 2014. Other studies reported that maize and wheat consume relatively more water in the North than the South of China (Tian et al., 2019; Wang et al., 2019). Developing water-saving irrigation has become an important way to alleviate the prominent contradiction between water resources utilization and grain production in China. According to NBSC (2021), the area of water-saving irrigation projects in China in 2019 was 37 million ha, including 7 million ha for micro irrigation. Therefore, micro irrigation does apply to food crops in China despite the limited irrigated area. For instance, in Xinjiang province, the area of micro irrigated maize and wheat was 0.033 million ha in 2009 (CIDDC, 2022), of which the wheat area dominated at up to 0.031 million ha (Wang et al., 2011). Meanwhile, some scholars are conducting research on micro irrigated maize (Bai and Gao, 2021; Guo et al., 2021) and wheat (Li et al., 2021; Zain et al., 2021) in China, especially in the North. Therefore, the

water consumption rates of these staple crops under future climate change scenarios with different irrigation techniques should
be closely monitored to ensure water supply and food crop production security in China and worldwide. Compared to existing
literatures on evaluation of WFs of crop production under climate change scenarios (e.g., Karandish et al., 2022), the
innovations of the current research are embodied in two points. The present study clarifies large-scale spatiotemporal responses
of WF to future climate change scenarios under different irrigation techniques for the first time. This analysis is also the first
to explore the large-scale future changes in WF benchmarks under different irrigation techniques.

**Table 1.** Inventory of global climate models (GCMs) used in the current study.

| GCM | Institute | Reference | Type |
|---|---|---|---|
| CCCMA-CanESM2 | Canadian Centre for Climate Modelling and Analysis | Arora et al. (2011); von Salzen et al. (2013) | Wet |
| CESM1-CAM5 | National Science Foundation, Department of Energy, National Center for Atmospheric Research | Hurrell et al. (2013) | |
| GFDL-CM3 | NOAA Geophysical Fluid Dynamics Laboratory | Delworth et al. (2006); Donner et al. (2011) | |
| FIO-ESM | The First Institute of Oceanography, State Oceanic Administration, China | Qiao et al. (2013) | Dry |
| GISS-E2R | NASA Goddard Institute for Space Studies USA | Schmidt et al. (2006); Schmidt et al. (2014) | |
| IPSL-CM5A-MR | Institute Pierre Simon Laplace | Dufresne et al. (2013) | |


## 2 Method and data

### 2.1 Research set-up

We studied the spatiotemporal responses of blue and green WF and corresponding WF benchmarks for two crops (maize
and wheat) to future climate change under two climate change scenarios (RCP2.6 and RCP8.5) using four different growing
modes (rain-fed and furrow-, micro-, and sprinkler-irrigated). First, we determined the baseline year. Second, we considered
different growing modes to quantify WF and corresponding WF benchmarks of two crops in the baseline year and future year
levels under two climate change scenarios. Finally, the spatiotemporal responses of crop WF and corresponding WF
benchmarks to future climate change were analysed (Fig. 1).


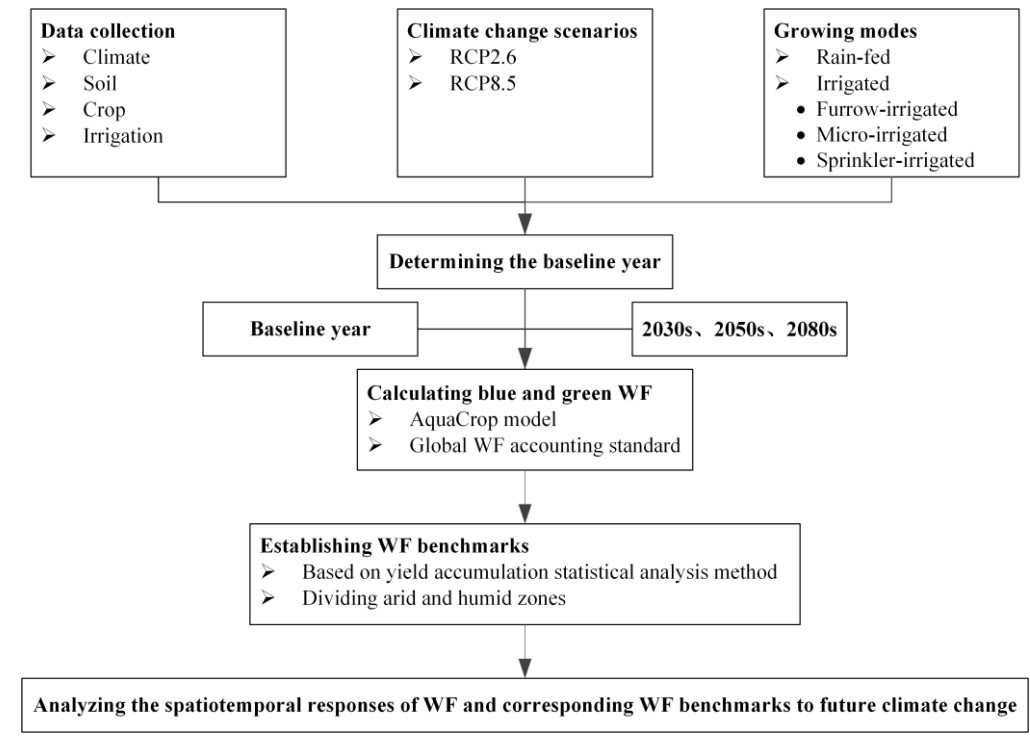


**Figure 1.** Flow chart for the study.

### 2.2 Determining the baseline year

Determining the baseline year is needed for a comparison between future and current conditions. Climate determines the annual variability of WF (Zhuo et al., 2014), and the baseline year should be determined when there is a relative balance between aridity and moisture. Hence, the aridity index (AI) was used here. Annual reference evapotranspiration ($ET_0$, mm) and precipitation (PR, mm) in China were calculated (Harris et al., 2014). Then, the AI was calculated, and climate change trends from 2000 to 2014 were analysed. The year 2013 was designated the baseline as its drought level was nearest the 15-year national average. The AI was calculated according to the method of Middleton and Thomas (1997):

$$AI = \frac{PR}{ET_0},$$ (1)

### 2.3 Water footprint per unit crop calculation

WF ($m^3$ $t^{-1}$) comprises blue WF ($WF_b$, $m^3$ $t^{-1}$) and green WF ($WF_g$, $m^3$ $t^{-1}$):

$$WF = WF_b + WF_g,$$ (2)

where $WF_b$ and $WF_g$ were calculated as the quotient of the blue ($CWU_b$, m³ ha⁻¹) and green ($CWU_g$, m³ ha⁻¹) components of crop water use (CWU, m³ ha⁻¹) and crop yield (Y, t ha⁻¹), respectively. $CWU_b$ and $CWU_g$ were equivalent to the cumulation of daily evapotranspiration (ET, mm d⁻¹) throughout the whole crop growth period (Hoekstra et al., 2011):

$$WF_b = \frac{CWU_b}{Y} = \frac{10 \times \sum_{d=1}^{lgp} ET_b}{Y} ,$$ (3)

$$WF_g = \frac{CWU_g}{Y} = \frac{10 \times \sum_{d=1}^{lgp} ET_g}{Y} ,$$ (4)

where $ET_b$ and $ET_g$ (mm) refer to the blue and green water evapotranspiration, respectively, and $lgp$ refers to the number of days of the crop growth period. The coefficient, 10, is a unit conversion factor, transforming the water depth of ET (mm) into the water amount per unit land area of CWU (m³ ha⁻¹).

The ET and Y per grid for each crop were simulated by the AquaCrop model based on the dynamic daily soil water balance (Mekonnen and Hoekstra, 2010):

$$S_{[t]} = S_{[t-1]} + PR_{[t]} + IRR_{[t]} + CR_{[t]} - ET_{[t]} - RO_{[t]} - DP_{[t]} ,$$ (5)

where $S_{[t]}$ and $S_{[t-1]}$ (mm) refer to the water content in soil when the day, t, ends and begins, respectively; $PR_{[t]}$ (mm) is the amount of precipitation on day, t; $IRR_{[t]}$ (mm) is the amount of water used for irrigation; $CR_{[t]}$ (mm) is the capillary rise to the crop root zone from the shallow groundwater; $RO_{[t]}$ (mm) is the water lost by surface runoff due to precipitation; and $DP_{[t]}$ (mm) is the water lost by deep percolation caused by excessive precipitation or irrigation. It was assumed that $CR_{[t]} = 0$ as the ground water depth was > 1 m (Allen et al., 1998). $RO_{[t]}$ was calculated using the Soil Conservation Service curve-number (CN) equation (USDA, 1964; Rallison, 1980):

$$RO_{[t]} = \frac{(PR_{[t]} - I_a)^2}{PR_{[t]} + S - I_a} ,$$ (6)

$$S = 254 \left( \frac{100}{CN} - 1 \right) ,$$ (7)

where $S$ (mm) is the potential maximum water storage, and $I_a$ (mm) is the initial amount of water loss before the runoff formation.

By tracking the daily flow of water in and out of the crop root zone, we separated the daily blue and green soil water balances (Zhuo et al., 2016b):

$$S_{b[t]} = S_{b[t-1]} + \left( PR_{[t]} + IRR_{[t]} - RO_{[t]} \right) \times \frac{IRR_{[t]}}{PR_{[t]} + IRR_{[t]}} - \left( DP_{[t]} + ET_{[t]} \right) \times \frac{S_{b[t-1]}}{S_{[t-1]}} ,$$ (8)

$$S_{g[t]} = S_{g[t-1]} + \left( PR_{[t]} + IRR_{[t]} - RO_{[t]} \right) \times \frac{PR_{[t]}}{PR_{[t]} + IRR_{[t]}} - \left( DP_{[t]} + ET_{[t]} \right) \times \frac{S_{g[t-1]}}{S_{[t-1]}} ,$$ (9)

where $S_{b[t]}$ and $S_{b[t-1]}$ (mm) are the blue water content in soil when the day, t, ends and begins, respectively; and $S_{g[t]}$ and $S_{g[t-1]}$ (mm) are the green water content in soil when the day, t, ends and begins, respectively. It is assumed that the initial soil water content before the crop growth period is green water.

In AquaCrop, the daily transpiration ($Tr_{[t]}$, mm) calculates the daily shoot biomass production (B, kg) using the normalised crop biomass water productivity (WP*, kg m⁻²) (Raes et al., 2017):

$$B=WP^* \times \sum \frac{Tr_{[t]}}{ET_{0[t]}} ,\tag{10}$$

where $WP^*$ is normalised to consider $CO_2$ concentration, reference evapotranspiration ($ET_0$), and crop classes (C3 or C4) so
that it is applicable to various locations and seasons. Water productivity remains constant for specific crops. Y, as the
harvestable portion of final B, is calculated by multiplying B with the adjusted reference Harvest Index ($HI_0$, %):

$$Y=f_{HI} \times HI_0 \times B ,\tag{11}$$

where $f_{HI}$ is a correction factor for $HI_0$. It considers the water and temperature stresses during the crop growth period. Being
consistent with the existing widely used scaling method (Mekonnen and Hoekstra, 2011; Zhuo et al., 2016b, 2016c, 2019;
Wang et al., 2019; Mialyk et al., 2022), the simulated Y per grid for each crop in 2013 was validated via scaling model
simulation outputs to correspond with the crop yield statistics data at the provincial level (NBSC, 2021). With the consistent
scaling factors for the Y simulation and crop parameters including the crop calendar, $WP^*$, $HI_0$, and the maximum root depth
which represent the existing agricultural production level, climate was the only variable for future scenario simulations.
In the simulation, different growing modes, namely rain-fed and three different irrigation techniques (furrow, micro, and
sprinkler irrigation), were considered. The irrigation schedule of three irrigation techniques in the model was the Generation
of Irrigation Schedule, namely the generation of an irrigation schedule by specifying a time and depth criterion for planning
or evaluating a potential irrigation strategy. The time criterion we used was Allowable depletion (%), namely the percentage
of the Readily Available soil Water (RAW) that can be depleted before irrigation water has to be applied. The depth criterion
we used was the Back to field capacity as the extra water on top of the amount of irrigation water required to bring the root
zone back to field capacity. The water quality was expressed by the Electrical conductivity (dS m$^{-1}$) of the irrigation water.
The soil surface wetted (%), an indicative value for the fraction of soil surface wetted, was used to select irrigation techniques.
Table 2 shows the parameters of three irrigation techniques (Raes et al., 2017). We can adjust the simulated ET and Y according
to the performance of the irrigation schedule.

**Table 2.** Parameters of three irrigation techniques.

| Irrigation technique | From day | Time criterion | Depth criterion | Water quality | Soil surface wetted |
|---|---|---|---|---|---|
| | | Allowable depletion | Back to field capacity | Electrical conductivity | |
| | | (%) | (+/- mm) | (dS m$^{-1}$) | (%) |
| Furrow | 1 | 50 | 10 | 1.5 | 80 |
| Micro | 1 | 20 | 10 | 0 | 40 |
| Sprinkler | 1 | 50 | 10 | 1.5 | 100 |


**2.4 Benchmarking consumptive WF in crop production**
Based on the work of Mekonnen and Hoekstra (2014), we ranked grid-level WF for each crop in ascending order of size
against the corresponding cumulative percentages of the total crop production. The annual WF of 20 % or 25 % of the producers
with the highest water productivity in China was set as the annual WF benchmark. The climate zones should be divided when
WF benchmarks are established (Zhuo et al., 2016a). To this end, the AI partitioned China into arid ($< 0.5$) and humid ($> 0.5$)
zones based on the annual $ET_0$ and PR from 2000 to 2014 at a 30-arc minute grid resolution (Harris et al., 2014) (Fig. 2).

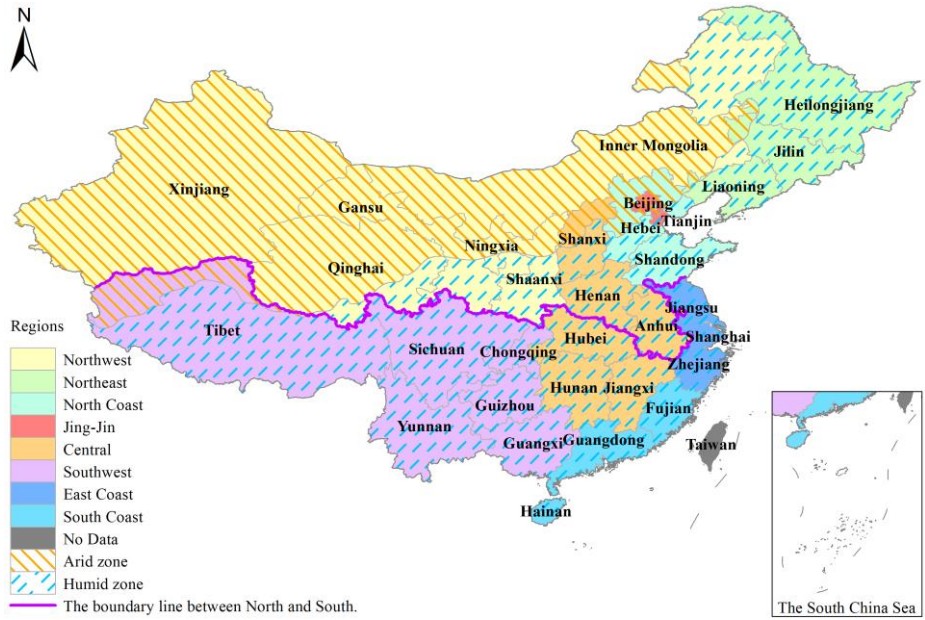

**Figure 2.** Regions and climate zones of mainland China.

**2.5 Data sources**
Monthly climate data, such as maximum (Tx), minimum air temperature (Tn), precipitation (PR), and reference
evapotranspiration ($ET_0$), from 2000 to 2014 at a resolution of 30-arc minute were derived from the CRU-TS 3.24 dataset
(Harris et al., 2014; CEDA, 2018). The mean annual atmospheric $CO_2$ concentration (ppm) from 2000 to 2014 was obtained
from the Mauna Loa Observatory, Hawaii, USA (NOAA, 2018). The downscaled outputs of six GCMs at a 5-arc minute grid
resolution in the 2030s, 2050s, and 2080s were obtained from the Climate Change, Agriculture and Food Security (CCAFS)
database (Navarro-Racines et al., 2020; CCAFS, 2015). As the CCAFS database has no $ET_0$ data, we calculated $ET_0$ for each
climate scenario using temperature inputs via the FAO Penman-Monteith method with missing data as described by Allen et
al. (1998). The projected $CO_2$ concentrations under RCP2.6 and RCP8.5 were obtained from van Vuuren et al. (2007) and
Riahi et al. (2007), respectively. To make the model simulation more in line with the actual situation in China, we reset the
maximum root depth ($Z_x$) according to the FAO-56 recommendation (Allan et al., 1998). The FAO-56 recommended values
provide clear range of the $Z_x$ for each type of crops for typical climatic zones. In addition, we further combined the literature
research on maize and wheat in China to reset the $HI_0$ (Zhuo et al., 2016c). The other parameters used in AquaCrop were
derived from Raes et al. (2017). Soil texture data and soil water capacity data at a 5-arc minute grid resolution were acquired
from the ISRIC Soil and Terrain database (Dijkshoorn et al., 2008) and ISRIC-WISE dataset (Batjes, 2012), respectively. The
planted areas for each irrigated or rain-fed crop at a 5-arc minute grid resolution were acquired from the MIRCA2000 dataset
(Portmann et al., 2010). We divided these planted areas into different parts subjected to various irrigation techniques using
statistical yearbook data (NBSC, 2021). Provincial-level crop yield statistics data were procured from the National Bureau of
Statistics of China (NBSC, 2021).
**3 Results**
**3.1 Future climate change trends in maize and wheat planted areas**
In the baseline year 2013, the average annual reference evapotranspiration ($ET_0$) and precipitation (PR) in the planted
areas of two crops were 941 mm and 727 mm, respectively. Compared with the baseline level of 2013, the average annual $ET_0$
and PR in the planted areas of two crops will both increase under two RCPs, and the increase in $ET_0$ exceeded that of PR. $ET_0$
will increase by 17 % and 29 % under RCP2.6 and RCP8.5, respectively, until the 2080s. However, PR will increase by 8 %
and 14 %, respectively. The increases under RCP8.5 (18–29 % and 3–14 % for $ET_0$ and PR, respectively) were much higher
than those under RCP2.6 (16–17 % and 4–8 % for $ET_0$ and PR, respectively). Climate change will be relatively more intense
under RCP8.5. The increases in $ET_0$ were concentrated from April to August (14–39 mm). The increases in PR were
concentrated between June and August (8–20 mm and 12–28 mm, respectively). However, PR will decline in May, July,
November, and December, and it will decline more in May ($\leq$ 9 mm until the 2030s) (Fig. 3a, b). Water and heat resources
were unevenly distributed in the planted areas of the two crops in 2013. $ET_0$ was relatively higher in East Coast and North
China. PR distribution was comparatively higher in the South and lower in the North (Fig. S4). Compared with 2013, $ET_0$ and
PR for the most heavily planted areas will increase under both scenarios until the 2080s. The areas with a relatively greater
increase in $ET_0$ were distributed mainly in Southwest and Northeast (Fig. 3c, e), and PR increased relatively faster in Northwest
and Jing-Jin (Fig. 3d, f). $ET_0$ decreased mainly in Xinjiang and Inner Mongolia (Fig. 3c, e), and PR decreased mainly in
Xinjiang, Tibet, Northeast, and South Coast (Fig. 3d, f). However, the areas where $ET_0$ decreased were 86–94 % smaller than
those where PR decreased.

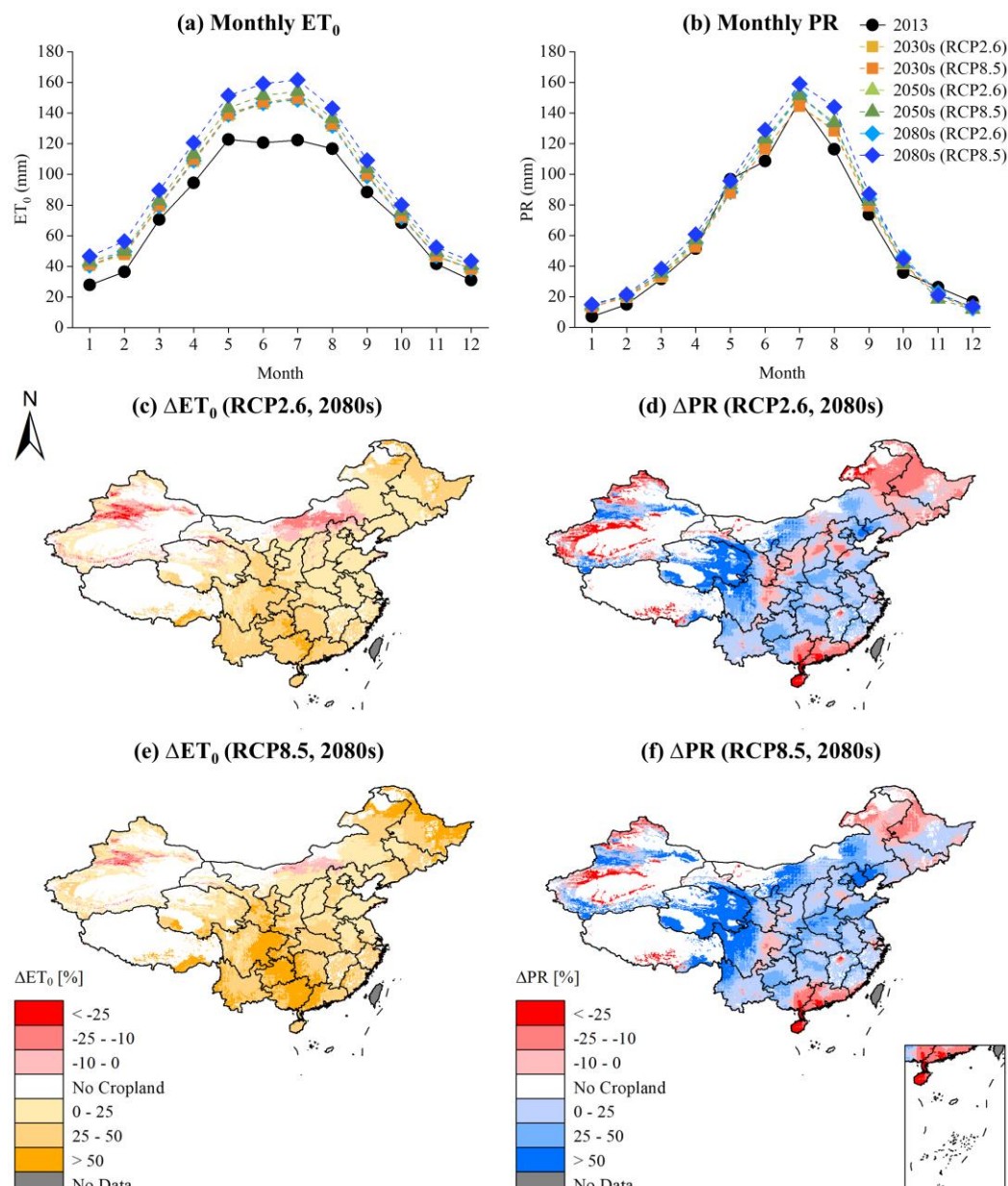

**Figure 3.** Future climate projections for the maize and wheat planted zones in China.

## 3.2 WF distribution in the baseline year 2013

The national average WF for wheat (1,008 m$^3$ t$^{-1}$) was higher than that for maize (813 m$^3$ t$^{-1}$) in the baseline year 2013. The corresponding blue WF proportions were 37 % and 20 %, respectively. The reason for this discrepancy is that maize is a C4 crop while wheat is a C3 crop. C4 crops have a relatively higher $CO_2$ fixation efficiency and faster photosynthetic rate than

C3 crops. Hence, maize can accumulate comparatively more yield than wheat under the same water consumption condition
(Wang et al., 2012). Figure 4 shows that the high $WF_g$ value was mainly distributed in areas with relatively greater precipitation
during crop growth, i.e., abundant green water resources. The main component of WF is $WF_g$; therefore, the high maize WF
was mainly distributed in Northwest (Fig. 4a), while the high wheat WF was mainly distributed in Southwest and South Coast
(Fig. 4b). Elevated $ET_0$ and insufficient precipitation can increase blue water consumption in food production. Thus, the high
$WF_b$ value was mainly distributed in areas with uneven water and heat resource distributions during crop growth. The high
maize $WF_b$ was mainly distributed in Northwest and East Coast (Fig. 4c), while that of wheat was distributed mainly in North
China (Fig. 4d). In all grids, the proportions of $WF_b$ and $WF_g$ were up to 68 % (wheat in Xinjiang) (Table S2) and 98 % (maize
in Hainan) (Table S1), respectively.

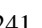


**Figure 4.** WF of maize and wheat in China in 2013.

A comparison of rain-fed and irrigation techniques demonstrated that the WF of maize and wheat under furrow and
sprinkler irrigation was higher than that under rain-fed in 2013. The WF of micro-irrigated crops was lower than that of rain-
fed crops. The WF of maize ($850\,m^3\,t^{-1}$) and wheat ($1,170\,m^3\,t^{-1}$) was highest under furrow and sprinkler irrigation, respectively.
For wheat under all three irrigation techniques, $WF_b$ was dominant (54–65 %). However, $WF_b$ for maize was only dominant
under micro irrigation (61 %). Micro-irrigated ($9.55\,t\,ha^{-1}$ for maize and $5.46\,t\,ha^{-1}$ for wheat) and rain-fed ($5.76\,t\,ha^{-1}$ for
maize and $4.51\,t\,ha^{-1}$ for wheat) crops had the highest and lowest yield, respectively, in 2013. The responses of maize yield to
rain-fed and various irrigation techniques were stronger than those of wheat yield (Fig. 4e, f).
**3.3 Spatiotemporal responses of WF to future climate change**
Compared with the baseline year 2013 and at the national average level, maize WF will increase under both RCP2.6 and
RCP8.5, by 17 % and 13 %, respectively, until the 2080s. The WF of wheat will increase under RCP2.6 (by 12 % until the
2080s) but decrease by 12 % under RCP8.5 until the 2080s (Fig. 5a). The increases in $CO_2$ concentration and, by extension,
yield gain, will be lower under RCP2.6 than RCP8.5. During the same period, the increases in WF under RCP2.6 will be 1–
3 % higher for maize and 2–10 % higher for wheat than those under RCP8.5. There will be relatively smaller differences in
$CO_2$ concentration between climate scenarios of the 2030s (431 ppm under RCP2.6 and 449 ppm under RCP8.5). Thus, the
differences in WF between RCPs will be smaller before the 2030s and larger after the 2050s. The WF of irrigated wheat under
RCP8.5 will decline by 3 % until the 2050s and by 15 % until the 2080s. The increase in WF will be highest under rain-fed,
and the WF of rain-fed maize and wheat under RCP2.6 will increase by 19 % and 24 %, respectively, until the 2080s. By
contrast, the WF of irrigated maize and wheat under RCP2.6 will only increase by 13 % and 7 %, respectively, until the 2080s
(Fig. 5a). A comparison of the various irrigation techniques demonstrated that the WFs of wheat and maize respond differently
under the same scenario. The increase in WF amplitude for maize will be highest under furrow irrigation (14 % and 11 %
under RCP2.6 and RCP8.5 until the 2080s, respectively) and lowest under micro irrigation (5 % and 2 % under RCP2.6 and
RCP8.5 until the 2080s, respectively). The WF of sprinkler-irrigated wheat under RCP8.5 will decline by 1 % until the 2030s.
The WF of wheat under micro irrigation had the highest increase (9 % until the 2080s under RCP2.6) and the lowest decrease
(14 % until the 2080s under RCP8.5). The WF of wheat under sprinkler irrigation had the lowest increase (only 2 % until the
2080s under RCP2.6) and the highest decrease (19 % until the 2080s under RCP8.5) (Fig. 5b).

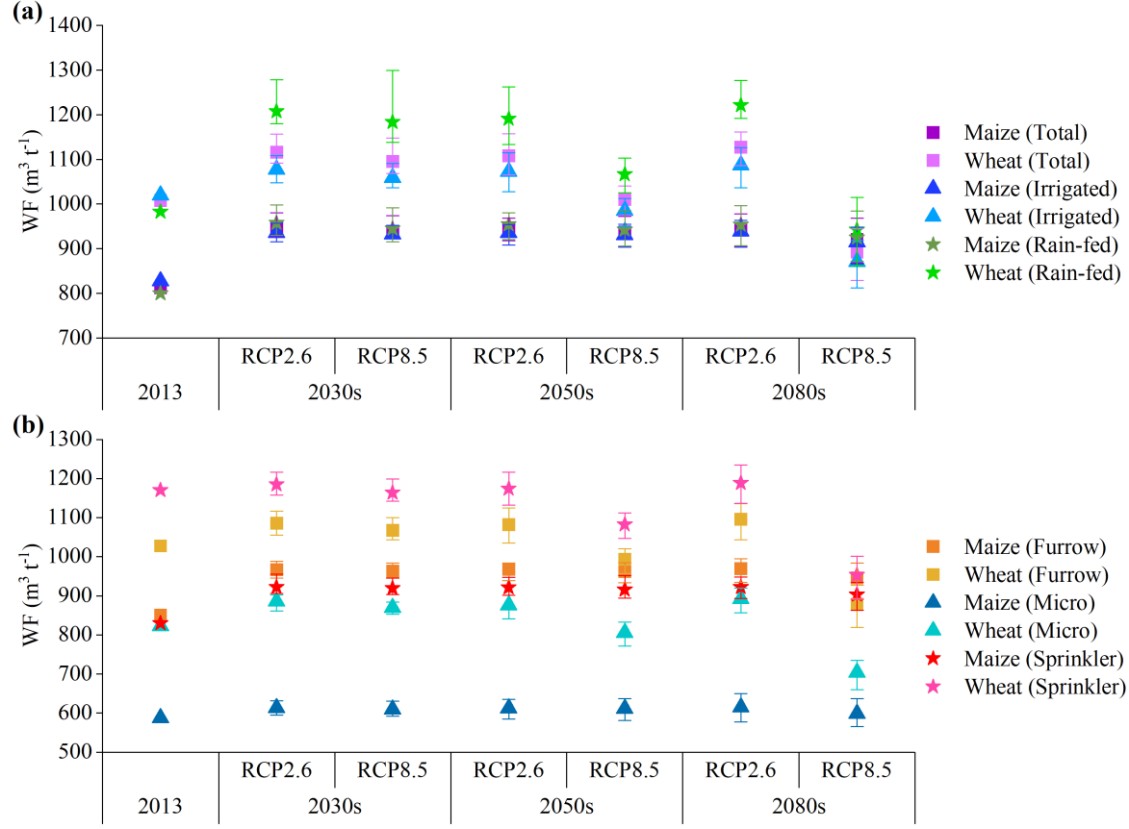

**Figure 5.** WF of maize and wheat in 2013 and future year levels under various climate change scenarios in China.

The spatial distribution of the relative changes in maize and wheat WF from 2013 to the 2080s showed regional differences. The WF will increase for 90–93 % of all areas planted with maize (Fig. 6a, b), and it will increase for 78 % of all areas planted with wheat under RCP2.6 (Fig. 6c) and decrease for 81 % of all areas planted with wheat under RCP8.5 (Fig. 6d). Increases in $ET_0$ lead to increases in WF, while decreases in PR lead to increases in $WF_b$ (Fig. S6). Hence, the regions with relatively greater increases in WF were mainly distributed where $ET_0$ strongly increased and PR slightly increased or even decreased. In Yunnan, maize WF increased by 44 % and 38 % under RCP2.6 and RCP8.5, respectively. In Guangxi, wheat WF increased by 50 % and 16 % under RCP2.6 and RCP8.5, respectively (Table S5). Comparison of rain-fed and various irrigation techniques revealed that the WF of each crop responded uniquely to latitudinal and longitudinal climate change under the same scenario. The responses of maize WF to climate change with latitude were relatively consistent. It increased by 27–43 % at 19–26 °N and ~51 °N latitude and decreased at ~44 °N latitude. By contrast, the responses of WF for rain-fed maize were more sensitive at ~40 °N and ~52 °N latitude. The responses of maize WF vary widely within 74–100 °E longitude. The WF of maize under rain-fed and furrow and sprinkler irrigation declined at 74–90 °E longitude. The increase in WF for maize under rain-fed at 93–98 °E longitude was 3–51 % higher than the increase in WF for maize under furrow and

sprinkler irrigation. The WF of micro-irrigated maize decreased at 74–95 °E longitude (Fig. 6a, b). The responses of wheat
WF to climate change with latitude and longitude were relatively consistent. However, in certain areas, there were large
differences in wheat WF between rain-fed and the three irrigation techniques. The WF of wheat under rain-fed decreased at
74–80 °E longitude and by more than the WF of wheat under the three irrigation techniques at the same longitude range. The
increases in the WF of wheat under rain-fed at ~93 °E and ~122 °E longitude and ~22 °N latitude were significantly higher
than the increases in WF of wheat under the three irrigation techniques (Fig. 6c, d).

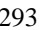

**Figure 6.** Spatial distributions in relative changes Δ (%) in WF (bottom left panel) with longitudinal (top panel) and latitudinal (right

panel) changes under different irrigation techniques applied to both crops under two scenarios from 2013 to the 2080s.


WF is determined by both crop yield (Y) and crop water use (CWU). We compared the relationships between the relative changes in WF ($\Delta$WF) and corresponding Y ($\Delta$Y) and CWU ($\Delta$CWU) (Fig. 7). The $\Delta$WF of maize and wheat under future climate change scenarios was inversely proportional to $\Delta$Y and directly proportional to $\Delta$CWU. Nevertheless, $\Delta$WF was relatively more sensitive to $\Delta$Y. When $\Delta$Y was 25 %, $\Delta$WF of wheat under RCP2.6 and maize was approximately -25 %, while $\Delta$WF of wheat under RCP8.5 was approximately -10 %. When $\Delta$CWU was 25 %, $\Delta$WF of wheat under RCP2.6 and maize was ~20 %, while $\Delta$WF of wheat under RCP8.5 was approximately -8 % (Fig. 7a, b). The responses of $\Delta$WF of maize were more sensitive to $\Delta$Y and $\Delta$CWU than those of wheat. The responses of $\Delta$WF of maize and wheat under RCP2.6 were more sensitive to $\Delta$Y and $\Delta$CWU than those under RCP8.5. Comparison of rain-fed and various irrigation techniques revealed that the correlation between $\Delta$WF and $\Delta$Y was stronger for rain-fed crops. For rain-fed maize, $R^2$ can reach 0.55 (Fig. 7a). $\Delta$WF and $\Delta$CWU were strongly correlated for irrigated crops, and $\Delta$WF and $\Delta$CWU were especially strongly correlated for crops under micro irrigation ($R^2$ can reach 0.98 for wheat) (Fig. 7b). We also determined the relationship between $\Delta$WF$_b$ and $\Delta$CWU$_b$ was similar but more significant than that between $\Delta$WF and $\Delta$CWU (Fig. 7c).

310

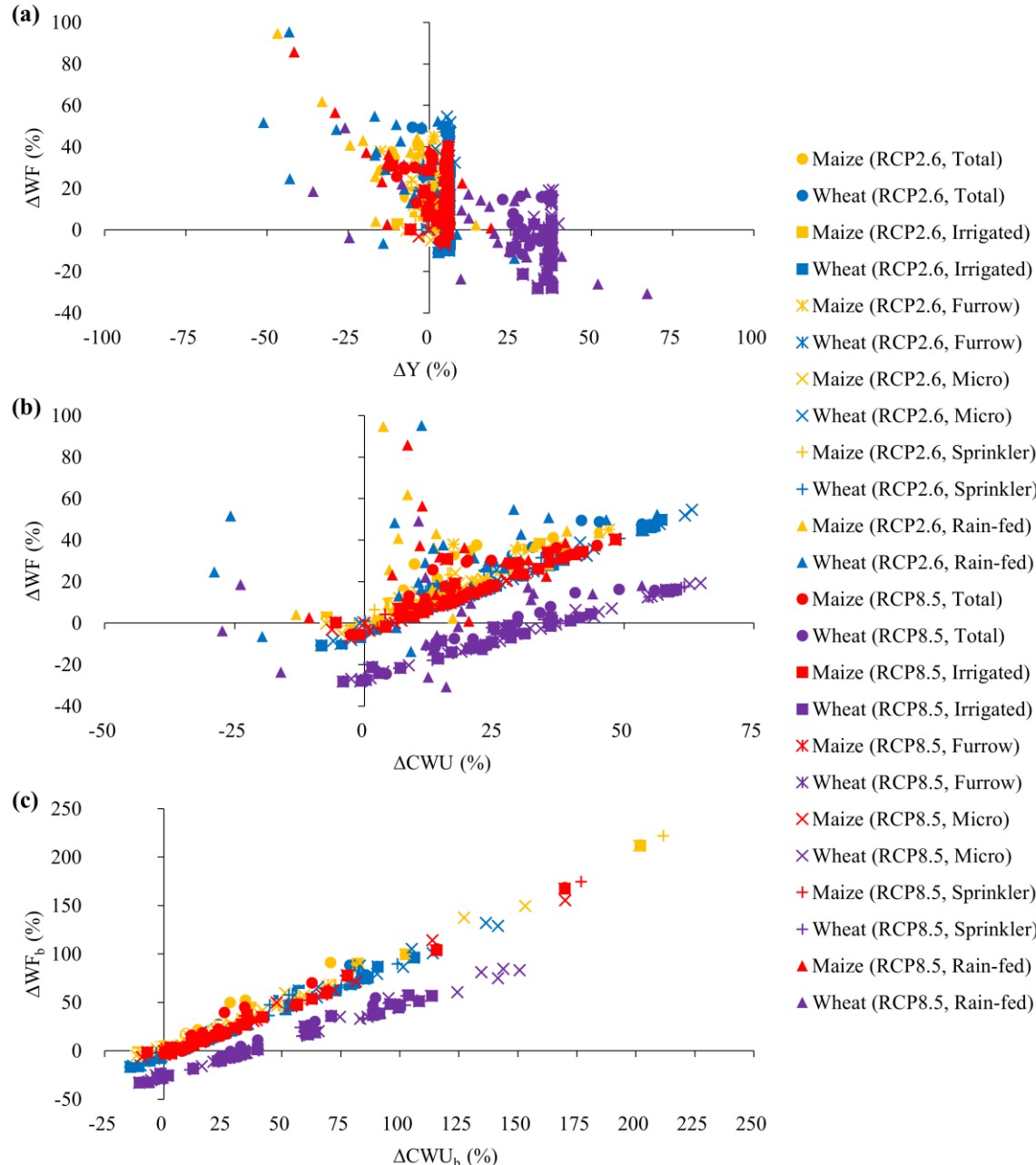

311

**Figure 7.** Relationships between relative changes Δ (%) in **(a)** Y and corresponding WF, **(b)** CWU and corresponding WF, and **(c)** CWU$_b$
and corresponding WF$_b$ of two crops under RCP2.6 and RCP8.5 from 2013 to the 2080s.

314

**3.4 Spatiotemporal WF benchmarks responses to climate change**

Table 3 shows the WF benchmarks of maize and wheat among various irrigation techniques and climate zones in 2013 and future year levels. The WF benchmarks of maize and wheat in the humid zone were 13–32 % higher than those in the arid zone, which is similar to results obtained by Wang et al. (2019). In the same climate zone, WF benchmarks of wheat were generally 2–35 % higher than those of maize. However, in the humid zone, the WF benchmark for the 25th production percentile of maize was 3 % higher than that of wheat under RCP8.5 in the 2080s. In the arid zone, WF benchmarks of rain-fed maize were 13–34 % higher than those of irrigated maize. In the humid zone of the future, WF benchmarks of rain-fed wheat were 2–7 % higher than those of irrigated wheat. In general, WF benchmarks of sprinkler-irrigated crops were higher, while those of micro-irrigated crops were lower. The differences in WF benchmarks among various irrigation techniques were more significant in the arid zone. WF benchmarks of the crops under micro irrigation were 30–38 % lower than those under sprinkler irrigation in the arid zone. The difference in the humid zone was only 8–14 %, which is also consistent with the study by Wang et al. (2019). In the humid zone, however, WF benchmarks of maize under furrow irrigation were 7–21 % higher than those under sprinkler irrigation.

**Table 3.** WF benchmarks ($m^3$ $t^{-1}$) of maize and wheat for different climate zones in 2013 and future year levels under two climate change scenarios in China.

| Climate zones | Crop | Type | WF ($m^3$ $t^{-1}$) at different production percentile[*] | | | | | |
| | | | 20th | | | 25th | | |
| | | | 2013 | RCP2.6 | RCP8.5 | 2013 | RCP2.6 | RCP8.5 |
|---|---|---|---|---|---|---|---|---|
| Arid | Maize | Total | 601 | (577, 576, 580) | (589, 584, 566) | 623 | (661, 658, 655) | (655, 652, 634) |
| | | Irrigated | 522 | (505, 504, 506) | (503, 503, 496) | 548 | (508, 507, 511) | (507, 509, 501) |
| | | Furrow | 618 | (658, 658, 658) | (654, 654, 642) | 654 | (693, 693, 691) | (689, 687, 674) |
| | | Micro | 466 | (455, 454, 456) | (456, 454, 440) | 477 | (459, 458, 460) | (458, 460, 446) |
| | | Sprinkler | 700 | (727, 725, 723) | (722, 719, 708) | 706 | (729, 729, 726) | (724, 721, 710) |
| | | Rain-fed | 599 | (661, 661, 662) | (652, 649, 630) | 618 | (682, 679, 671) | (672, 667, 652) |
| | Wheat | Total | 753 | (776, 764, 781) | (765, 707, 620) | 768 | (829, 816, 828) | (809, 756, 666) |
| | | Irrigated | 754 | (776, 764, 781) | (765, 707, 620) | 768 | (830, 816, 829) | (810, 757, 666) |
| | | Furrow | 830 | (850, 840, 850) | (830, 774, 680) | 940 | (885, 875, 887) | (868, 809, 712) |
| | | Micro | 648 | (701, 690, 705) | (694, 643, 562) | 670 | (717, 705, 721) | (707, 654, 572) |
| | | Sprinkler | 1020 | (1003, 998, 1007) | (989, 920, 811) | 1032 | (1034, 1028, 1038) | (1019, 948, 837) |
| | | Rain-fed | 692 | (743, 734, 753) | (729, 692, 618) | 692 | (790, 772, 791) | (769, 737, 653) |
| Humid | Maize | Total | 680 | (761, 754, 752) | (756, 752, 739) | 718 | (813, 807, 807) | (809, 806, 785) |
| | | Irrigated | 743 | (905, 905, 908) | (902, 900, 881) | 782 | (939, 939, 944) | (937, 936, 916) |
| | | Furrow | 762 | (925, 926, 930) | (921, 921, 901) | 801 | (943, 942, 948) | (940, 939, 919) |
| | | Micro | 649 | (709, 704, 707) | (694, 696, 683) | 660 | (734, 726, 732) | (721, 726, 708) |
| | | Sprinkler | 713 | (770, 771, 768) | (764, 762, 750) | 737 | (813, 814, 812) | (808, 806, 793) |
| | | Rain-fed | 631 | (712, 703, 707) | (710, 702, 678) | 656 | (744, 737, 737) | (740, 736, 716) |
| | Wheat | Total | 873 | (933, 932, 946) | (921, 851, 752) | 887 | (944, 942, 957) | (931, 860, 760) |

| | | | | | | |
|---|---|---|---|---|---|---|
| Irrigated | 887 | (914, 914, 924) | (900, 841, 744) | 897 | (925, 926, 937) | (912, 849, 752) |
| Furrow | 887 | (914, 914, 925) | (901, 841, 744) | 896 | (925, 927, 937) | (913, 849, 752) |
| Micro | 820 | (821, 826, 838) | (804, 753, 665) | 833 | (830, 839, 849) | (812, 759, 671) |
| Sprinkler | 933 | (949, 944, 955) | (936, 872, 770) | 946 | (958, 953, 964) | (944, 880, 777) |
| Rain-fed | 812 | (973, 958, 984) | (950, 863, 757) | 831 | (989, 973, 998) | (964, 877, 763) |

*The three numbers in brackets are the values of 2030s, 2050s and 2080s.

331

Compared with the baseline year, 2013, the changes in maize and wheat WF benchmarks under future climate change scenarios are similar to the changes in WF. However, the WF benchmark for the 20th production percentile of maize will decline by 2–6 % in the arid zone. WF benchmarks of wheat under RCP8.5 will decrease by 2–6 % and 13–18 % until the 2050s and the 2080s, respectively. The increasing range of the WF benchmark for the 25th production percentile of maize was 7–8 % higher in the humid zone than that in the arid zone. The increasing range of the WF benchmark for the 20th production percentile of wheat was 4–5 % higher in the humid zone than that in the arid zone. WF benchmarks of maize and wheat increased to a greater extent under RCP2.6 but decreased to a greater extent under RCP8.5. WF benchmarks of rain-fed crops increased more than those of irrigated crops in the same climate zone. Nevertheless, the increase in WF benchmarks was 7–11 % lower for rain-fed than irrigated maize in the humid zone. WF benchmarks of maize and wheat generally increased relatively more under furrow irrigation and comparatively less under sprinkler irrigation. However, under RCP2.6, the growth rate of the WF benchmark for the 20th production percentile of wheat was 5–6 % higher under micro irrigation than that under furrow irrigation in the arid zone. The increase in the WF benchmark for the 20th production percentile of wheat was 0.19–2 % higher under sprinkler irrigation than that under micro irrigation in the humid zone (Table 3).

Figure 8 shows the spatial distribution of the relative changes in the WF of maize and wheat compared with the benchmark for the 25th production percentile in 2013 and the 2080s. In 2013, the WF for 81 % and 79 % of the maize and wheat planted areas, respectively, was higher than its benchmark. The maize planted areas with WF below the benchmark were distributed mainly in Xinjiang in the arid zone and northeast Inner Mongolia in the humid zone (Fig. 8a). The wheat planted areas with WF below the benchmark were distributed mainly in Xinjiang in the arid zone and Qinghai (Fig. 8d). Under future climate change scenarios, the maize and wheat planted areas with the WF below the benchmark will slightly decrease in the 2080s. These areas are mainly distributed in Heilongjiang, Tibet, southern Gansu, and Sichuan in the humid zone for maize; and Henan and Tibet in the humid zone and Qinghai for wheat. This is because that the annual $ET_0$ will increase relatively faster in Heilongjiang and Tibet, which will lead to a greater increase in $WF_b$. The annual PR in other regions will significantly increase, which will result in a greater increase in $WF_g$. Maize and wheat planted areas under RCP8.5 with WF below the benchmark will decrease by 5 % and 4 %, respectively, until the 2080s.

356

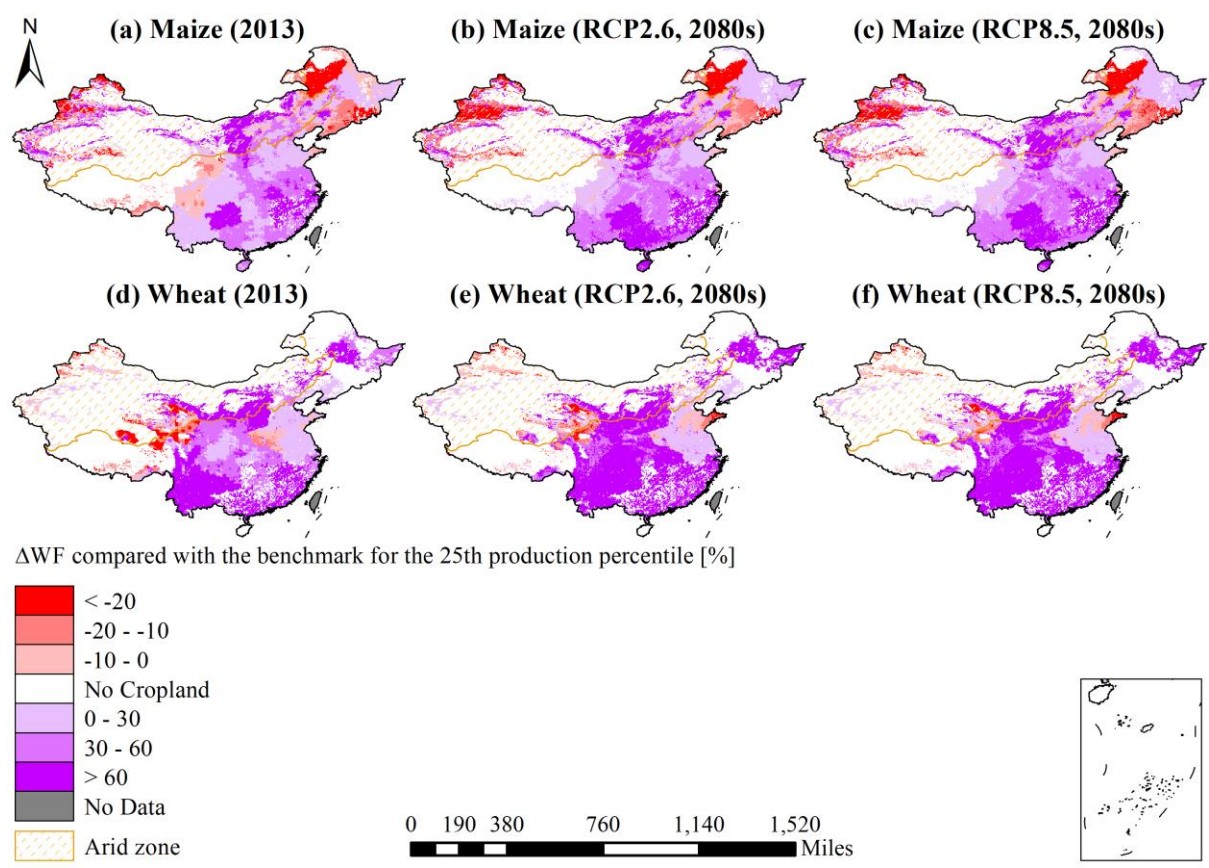

**Figure 8.** Relative changes Δ (%) in the WF of maize and wheat compared with the benchmark for the 25th production percentile in 2013 and the 2080s under RCP2.6 and RCP8.5 in different climate zones of China.

**3.5 Discussion**

This study analysed and compared the WF and WF benchmarks responses of wheat and maize under rain-fed and various irrigation conditions and forecasted their responses to future climate change scenarios in China. Under the background that the annual $ET_0$ and PR will both increase but $ET_0$ will increase faster, maize WF will increase under both RCP2.6 and RCP8.5. Wheat WF will increase under RCP2.6 but decrease under RCP8.5 until the 2080s. Rain-fed crops had higher ranges of increasing WF, which is consistent with Rosa et al. (2020). The increasing ranges of maize and wheat WF were lowest under micro irrigation and sprinkler irrigation, respectively. Therefore, the implementation of water-saving irrigation techniques (micro and sprinkler irrigation) may help mitigate the adverse effects of future climate change on agriculture, which is in line with Dai et al. (2020). Under future climate change, WF benchmarks will be modified in a manner resembling that for WF. However, the former changes will not be as significant as the latter in the same area.

In 2013, the WF of maize was lower than that of wheat. Nevertheless, maize WF is expected to increase more rapidly than wheat WF under future climate change scenarios. C4 crops such as maize have higher photosynthetic rates than C3 crops such as wheat. However, C4 crops are less sensitive to elevated atmospheric $CO_2$ than C3 crops (Bowes, 1993). Hence, while maize yield is higher than wheat yield, the former increases less than the latter. We compared current results against those of previous studies in Table 4. The differences we determined for the relative changes in maize and wheat WF between years and RCPs resembled those reported by Zhuo et al. (2016d). However, these authors also considered other factors, such as harvested crop area, technology, diet, and population, that could partially offset the adverse effects of future climate change. Therefore, maize and wheat WF will decline in the future according to Zhuo et al. (2016d). Fader et al. (2010) studied relative global-scale changes in maize WF for 2050. Their analysis was conducted in the opposite direction of that of the present study on China. Moreover, the two studies differed in terms of climate scenario, research area, and crop model. Winter wheat WF in Germany and Italy will decline by 2050 according to Garofalo et al. (2019). Nevertheless, our research showed that winter wheat WF will increase in China by 2050. The crop water use in Germany and Italy changes more smaller than that in China. However, our observed differences in the relative changes in WF between RCPs were consistent with those of Garofalo et al. (2019); namely, under RCP8.5, WF will either decrease more or increase less.

**Table 4.** Comparison of the results between current and previous studies.

| Reference | Year | Study case | Scenario | Relative changes in WF (%) |
|---|---|---|---|---|
| Zhuo et al. (2016d) | 2030 | China Maize | RCP2.6 / RCP8.5 | -38–-32 / -10–0 |
| | | China Wheat | | -25–-17 / -20–-11 |
| | 2050 | China Maize | | -51–-43 / -22–-8 |
| | | China Wheat | | -36–-27 / -38–-27 |
| Current study | 2030s (2020–2049) | China Maize | RCP2.6 / RCP8.5 | 17 / 16 |
| | | China Wheat | | 11 / 9 |
| | 2050s (2040–2069) | China Maize | | 16 / 15 |
| | | China Wheat | | 10 / 0.20 |
| Fader et al. (2010) | 2041–2070 | Global Maize | SRES A2 | -0.44–-0.35 |
| Current study | 2050s (2040–2069) | China Maize | RCP2.6 / RCP8.5 | 16 / 15 |
| Garofalo et al. (2019) | 2050 | Germany Winter wheat | RCP4.5 / RCP8.5 | -24 / -26 |
| | | Italy Winter wheat | | -5 / -6 |
| Current study | 2050s (2040–2069) | China Winter wheat | RCP2.6 / RCP8.5 | 10 / 0.60 |

In the future, the spatial distributions of maize and wheat WF will change considerably. By contrast, the spatial distributions of WF benchmarks will negligibly change. This phenomenon is comparatively more pronounced in the area with limited agricultural development. In 2013, Guizhou and Guangxi had the highest maize and wheat WF (1,317 m$^3$ t$^{-1}$ and 3,720 m$^3$ t$^{-1}$, respectively) (Table S1, S2). In the humid zone, maize WF in Guizhou and wheat WF in Guangxi will increase by 37 % and 50 %, respectively, under RCP2.6 and by 33 % and 16 %, respectively, under RCP8.5 until the 2080s (Table S5). Nevertheless, the WF benchmarks for the 25th production percentile of maize and wheat in the humid zone will only increase by 12 % and 8 %, respectively, under RCP2.6 and increase by 9 % and decrease by 14 %, respectively, under RCP8.5. These

areas will nonetheless have great potential for agricultural water conservation in the future. If maize and wheat WF in various regions of China can be reduced to the benchmark for the 25th production percentile, the total CWU can be reduced by 45–66 billion $m^3$ (~14–17 %). Rain-fed agriculture can save 27–40 billion $m^3$ (~18–22 %), water which is more than that conserved by irrigation. In irrigated agriculture, furrow irrigation has a comparatively high water-saving potential (17–22 billion $m^3$; ~11–12 %). To optimise the agricultural water-saving potential in China, we must either reduce WF or prevent it from increasing, either by enhancing crop yield or decreasing CWU. However, this goal can only be realised with the support of relevant policies and management practices. The annual PR is relatively low, and the $ET_0$ is relatively high in North China. Shortage of water for agriculture is a major bottleneck in the development of local agriculture there. However, furrow irrigation is mainly applied in these areas (Fig. S3). Hence, irrigation water use efficiency is low and $WF_b$ is high. High-efficiency, water-saving micro irrigation, and sprinkler irrigation could replace furrow irrigation in these areas so that CWU and WF decrease. The planted areas in the South have abundant precipitation but limited distribution (Fig. S2) and high WF (Fig. 4a, b). WF can be mitigated by implementing ground cover techniques (ex. straw return, mulch) to reduce soil evaporation and by improving farmer skills. WF can also be reduced by optimizing the structure of crop planting. Crops and varieties best adapted to local climate conditions and climate change can lower irrigation requirements and reduce WF.

To make climate models comparable and promote their development, The World Climate Research Program (WCRP) has developed and promoted the CMIP since 1995 (Meehl et al., 1997, 2000). Its current iteration is CMIP Phase 6 (CMIP6), which will be used in the forthcoming Intergovernmental Panel on Climate Change's Sixth Assessment Report (IPCC AR6). GCMs and their associated research results based on CMIP5 provided vital support for IPCC's Fifth Assessment Report (IPCC AR5). CMIP5 proposed four RCP scenarios (RCP2.6, RCP4.5, RCP6.0, and RCP8.5) by considering greenhouse gas (GHG) emissions and concentrations, atmospheric pollutant concentrations, and land use in the 21[st] century (Moss et al., 2008). However, no specific socio-economic assumptions were made. The Scenario Model Intercomparison Project (ScenarioMIP), as the primary activity within CMIP6, will provide a series of new climate scenarios that consider social factors related to climate change adaptation and impacts. They will be based on the combined application of shared socioeconomic pathways (SSPs) and RCPs and will compensate for the limitations of the RCPs in CMIP5 (O'Neill et al., 2016). The climate models in CMIP5 and CMIP6 can both effectively simulate changes in potential evapotranspiration (Liu et al., 2020) and precipitation (Müller et al., 2021) in most parts of the world. Müller et al. (2021) reported that CMIP5 and CMIP6 simulate increasing trends in temperature in a similar fashion. Nevertheless, the simulation generated by CMIP6 is higher than that by CMIP5. Notwithstanding, CMIP5 and CMIP6 are reasonably consistent and similar in terms of their abilities to predict future climate changes. This study focused on the responses of crop production to future climate change. It mainly considered the influences of GHG emission- and concentration-driven climate change and excluded the influences of alterations in socioeconomic development. Therefore, we implemented CMIP5 in our current research.

Three are two methods of establishing WF benchmarks (Hoekstra, 2013). Method 1 is based on yield accumulation statistical analysis. Due to the variability of WFs found across regions and among producers within a region, for each crop, we can select the WF of 20 % or 25 % of the producers with the highest water productivity as the WF benchmark (Mekonnen and

Hoekstra, 2014). Method 2 is based on the available optimal technique analysis. We can compare the WFs at each location under different agricultural management practices and take the WF associated with optimal practice, which results in the smallest WF, as the WF benchmark (Chukalla et al., 2015). Both methods establish WF benchmarks based on the maximum reasonable water consumption in each step of the product's supply chain (Hoekstra, 2014). Method 1 is suitable for large-scale application. The differences in environmental conditions (such as climate) and development conditions should be considered comprehensively (Mekonnen and Hoekstra, 2014; Zhuo et al., 2016a). The drawback of Method 1 is that no matter what spatial scope one takes in grouping producers, within that scope there will still be variability from place to place even if the differences in regional environmental and development conditions are taken into account (Schyns et al., 2022). Method 2 is suitable for smaller scale and overcomes this drawback of Method 1 to some extent. The Method 2's drawback is that it has the higher requirements on the setting and simulation of different agricultural management practices. We mainly want to explore the response of large-scale WF to future climate change under specific irrigation technique, that is, each irrigation technique has its corresponding WF benchmarks. And only one agricultural management practice, that is irrigation, is considered here. Therefore, we choose Method 1. A combination of methods should be established. If conditions permit, we strongly recommend that Method 1 and Method 2 are combined to establish small-scale WF benchmarks. Different agricultural management practices, such as irrigation, mulching techniques and so on, can be combined to further determine WF benchmarks.

The sources of uncertainty in research on the responses of crop production to climate change include GCMs, climate scenarios, crop models, and their interactions (Wang et al., 2020). Semenov and Stratonovitch (2010) proposed that the use of multiple GCMs can reduce the uncertainty associated with them. We selected three GCMs each for wet and dry climate outputs to encompass a broad climate prediction scenario. To objectively and comprehensively project the future climate change trends of China, we selected two extreme RCPs, namely, RCP2.6 and RCP8.5. Wang et al. (2020) suggested that crop models are the main source of uncertainty in predicting wheat yield in China under future climate change. The application of various crop models and parameter settings inevitably lead to different yield forecasts (Asseng et al., 2013). Hence, the use of AquaCrop alone may introduce uncertainty into WF forecasting.

The present study had certain limitations in terms of the assumptions it made for the simulation. First, we assumed that the crop parameters (such as planting calendar, $HI_0$, and $Zx$) for each crop under the identical growing mode (irrigated or rainfed) were constant on a spatiotemporal scale. Yoon and Choi (2020) proposed that future increases in temperature and precipitation might shorten the crop growth period. Xiao et al. (2020) indicated that the winter wheat and summer maize growing periods will be lengthened and shortened, respectively, under future climate change. However, we did not consider future changes in the crop growth period. Second, we assumed a constant soil surface moisture rate for each grid under the various irrigation techniques. Third, it was assumed that the observed changes in the planted areas in 2013 were based on the 2000 raster database, and we ignored the migration of planted areas. Finally, we assumed that the maize and wheat planted areas will not change in the future and would remain consistent with baseline year 2013. Thus, we did not consider future development of cultivated lands.

The core content of this study was to quantify the responses of maize and wheat WF and WF benchmarks to future climate
change under various irrigation techniques. Future research must improve the accuracy of the crop model simulation and
reduce the uncertainty of climate prediction associated with using different GCMs. Moreover, this study only considered future
climate change scenarios. Future investigations should also consider the influence of changes in technological development,
land use, growing modes, and so on.
**4 Conclusions**
This study explored the responses of maize and wheat WF accounting and benchmarking to future climate change in
China. The crops were subjected to various irrigation techniques. The year 2013 was the baseline, and WF and its benchmarks
were quantified for each crop under rain-fed and irrigation (furrow, micro, and sprinkler) management techniques in the 2030s,
2050s, and 2080s under RCP2.6 and RCP8.5 at a 5-arc grid scale. The AquaCrop model with the outputs of six GCMs in
CMIP5 as its input data was used to simulate the WF of maize and wheat. The results show that: (1) Compared with 2013, the
annual $ET_0$ and PR in the maize and wheat planted areas of China will both increase; however, the former will increase faster
than the latter. (2) Maize WF will increase under both RCP2.6 and RCP8.5 by 17 % and 13 %, respectively, until the 2080s.
Wheat WF will increase under RCP2.6 (by 12 % until the 2080s) but decrease by 12 % under RCP8.5 until the 2080s. Rain-
fed crops were more vulnerable to the adverse impacts of future climate change, and their WF increased to a greater extent
than that of irrigated crops. Micro irrigation and sprinkler irrigation resulted in the lowest increases in WF for maize and wheat,
respectively. Hence, these water-saving irrigation practices effectively mitigated the negative impact of climate change. (3)
Within different climate zones and under various irrigation techniques, there will be significant differences in the responses of
WF benchmarks to future climate change. The changes in WF and its benchmarks will be similar in response to future climate
change. The rate of increase in WF benchmarks for sprinkler-irrigated crops will generally be lower than those for rain-fed,
micro-irrigated, and furrow-irrigated crops within the same climate zone. However, the change in the spatial distribution of
WF benchmarks will not be as significant as that of WF itself. Moreover, this difference will be more pronounced in the region
with low agricultural development. Additionally, this study also demonstrated that the agricultural water in China still has
substantial water-saving potential and can be effectively conserved.

**Data availability.** Data sources are listed in Sect. 2.5. Data generated in this paper are available by contacting La Zhuo.
**Competing interests.** The authors declare that they have no conflict of interest.

**Author contribution**

La Zhuo and Pute Wu designed the study. Zhiwei Yue and Xiangxiang Ji carried it out, and prepared the manuscript with contributions from all co-authors.

**Acknowledgements**

The study is financially supported by the Program for Cultivating Outstanding Talents on Agriculture, Ministry of Agriculture and Rural Affairs, People's Republic of China [13210321], and the National Natural Science Foundation of China Grants [51809215].

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
