# Peer review of "Spatiotemporal responses in crop water footprint and benchmark under different irrigation techniques to climate change scenarios in China"

_Hydrology and Earth System Sciences, 2021_

## Author Comment (AC1)

**Authors' responses to interactive comments on "Spatiotemporal responses in crop water footprint and benchmark under different irrigation techniques to climate change scenarios in China"**

Dear Referee #1,

Thank you very much for your valuable comments and suggestions on our manuscript. We have provided our responses directly below each of the comments.

Anonymous Referee #1

**General comments**

1. One could divide WFP benchmarking techniques into two methods. Method 1 compares the WFP of different producers (or grid cells) within the same region, ranks them and sets a benchmark based on some percentile. Method 2 compares the WFP at each location under different management practices and sets a benchmark based on best practices (those resulting in the smallest WF). Method 2 is for example applied in the studies by Chukalla et al. (https://doi.org/10.5194/hess-21-3507-2017 and https://doi.org/10.5194/hess-19-4877-2015). The drawback of method 1 is that no matter what spatial scope one takes in grouping producers, within that scope there will still be variability from place to place (section 4.3.2.1 in https://doi.org/10.1016/B978-0-12-822112-9.00006-0). Rainfall, for example, shows strong spatial variability over short distances, such that a few producers in a larger area simply had more favourable local circumstances. Therefore, one can always question the comparability of producers that operate in different locations and the WFPs they achieve. Method 2 overcomes this drawback. In this manuscript method

1 is applied, although different irrigation practices are simulated. What are your reasons for determining the benchmarks based on method 2? Why don't you determine the benchmarks (also) based on method 1? You seem to have the data/simulations for that.

**Response:** We deeply appreciate your valuable comment. Sure, we acknowledge that three are two methods of establishing WF benchmarks (Hoekstra, 2013). The reason why we choose Method 1 instead of Method 2 is that we mainly explore the responses of large-scale WFs for two grain crops to future climate changes under specific irrigation technique, that is, each irrigation technique has its corresponding WF benchmarks. If Method 2 was selected, what we concerned about was the responses of WF to future climate change under the optimal tillage plus irrigation techniques which would result in the smallest WF. It is inconsistent with the current research objectives. This is also relevant to your General comment 2. Method 2 has the higher requirements on the setting and simulation of different agricultural management practices. However, we focus on only one agricultural management practice here, i.e., irrigation. If we chose Method 2, the calibration of different agricultural management practices would be the key. The existing data cannot meet these requirements for such a large-scale study. Therefore, we choose Method 1 to determine WF benchmarks. And we also realized that the application of Method 1 does have some limitations. We will add the consideration to Section 3.5 Discussion on the choice of two WF benchmarking methods and different agricultural management practices in combination with your Specific comment 13. The content is as follows.

"Three are two methods of establishing WF benchmarks (Hoekstra, 2013). Method 1 is based on yield accumulation statistical analysis. Due to the variability of WFs found across regions and among producers within a region, for each crop, we can select the WF of 20 % or 25 % of the producers with the highest water productivity as the WF benchmark (Mekonnen and Hoekstra, 2014). Method 2 is based on the available optimal technique analysis. We can compare the WFs at each location under different

agricultural management practices and take the WF associated with optimal practice, which results in the smallest WF, as the WF benchmark (Chukalla et al., 2015). Both methods establish WF benchmarks based on the maximum reasonable water consumption in each step of the product's supply chain (Hoekstra, 2014). Method 1 is suitable for large-scale application. The differences in environmental conditions (such as climate) and development conditions should be considered comprehensively (Mekonnen and Hoekstra, 2014; Zhuo et al., 2016a). The drawback of Method 1 is that no matter what spatial scope one takes in grouping producers, within that scope there will still be variability from place to place even if the differences in regional environmental and development conditions are taken into account (Schyns et al., 2022). Method 2 is suitable for smaller scale and overcomes this drawback of Method 1 to some extent. While the Method 2's drawback is that it has the higher requirements on the setting and simulation of different agricultural management practices. We mainly want to explore the response of large-scale WF to future climate change under specific irrigation technique, that is, each irrigation technique has its corresponding WF benchmarks. And only one agricultural management practice, that is irrigation, is considered here. Therefore, we choose Method 1. If conditions permit, we strongly recommend that Method 1 and Method 2 be combined when establishing small-scale WF benchmarks. We can consider different agricultural management practices, such as irrigation, mulching techniques and so on. They can be combined to further determine WF benchmarks."

2. AquaCrop provides crop parameters sets for maize and wheat which are to some degree calibrated for the conditions of recent history. How do you make sure the model produces reliable results for ET and Y under climate change scenarios?

**Response:** We are very sorry for our unclear expression and use of incorrect words. To some extent, we guaranteed how evapotranspiration (ET) and yield (Y) will develop in the future at the current production level. First, we chose 2013, when the drought level

came closest to the 15-year national average drought level from 2000 to 2014, as the baseline year. Second, the simulated Y per grid for each crop in 2013 was calibrated via scaling model simulation outputs to accord with the crop yield statistics data at the provincial level (NBSC, 2021), which was consistent with the widely used calibration method (Mekonnen and Hoekstra, 2011; Zhuo et al., 2016b, 2016c, 2019; Wang et al., 2019; Mialyk et al., 2022). For sure, the calibrated Y corresponded to the simulated ET. The crop parameters in model represent the existing agricultural production level. Climate is the only variable for future scenario analysis.

3. Micro-irrigation results in the smallest WFP and largest Y (Figure 3). Yet how feasible (and profitable) is micro irrigation in maize and wheat production in practice? Is it commonly applied for these crops in some parts of the world? Or is micro-irrigation mostly used for cash-crops only? Some elaboration on this in the manuscript is needed to justify the research setup and to put the results into perspective.

**Response:** We appreciate your valuable suggestion. Due to the high cost and technical requirements, micro irrigation applications are limited and are mostly applied to cash crops. However, there is a serious shortage of water resources in some croplands in China. And the spatial and temporal distribution of water and soil resources is uneven. Developing water-saving irrigation has become an important way to alleviate the prominent contradiction between water resources utilization and grain production in China. According to NBSC (2021), the area of water-saving irrigation projects in China in 2019 was 37 million ha, including 7 million ha for micro irrigation. Therefore, micro irrigation does apply to food crops in China despite the limited irrigated area. For instance, in Xinjiang province, the area of micro irrigated maize and wheat has been 0.033 million ha in 2009 (CIDDC, 2022), of which wheat area accounted for the main component, up to 0.031 million ha (Wang et al., 2011). Meanwhile, some scholars are conducting research on micro irrigated maize (Bai and Gao, 2021; Guo et al., 2021) and wheat (Li et al., 2021; Zain et al., 2021) in China, especially in the North.

4. What assumption do you take in terms of irrigation strategy/scheduling? This needs to be added to the methods. And how does this affect your results? This is important to address in the discussion, preferably with some quantitative substantiation. The more irrigation events you have, the more effect you will see from moving to a more efficient irrigation application technology (from furrow to drip). So I suppose your outcomes in terms of WFP for different irrigation technologies are quite sensitive to the assumption for the irrigation trigger (x% of soil moisture depletion?) and amount (back to field capacity?).

**Response:** We are very sorry for our unclear expression and deeply appreciate your valuable comment. We realized that description of different irrigation techniques settings was missing in Section 2.3 (original 2.2) Water footprint per unit crop calculation. Therefore, the following will be added at the end of Section 2.3. And the relevant table will be placed in the supplementary material.

"In the simulation, we considered different planting modes, namely rain-fed and three different irrigation techniques (furrow, micro, and sprinkler irrigation). The irrigation schedule of three irrigation techniques in model was Generation of Irrigation Schedule, namely the generation of an irrigation schedule by specifying a time and depth criterion for planning or evaluating a potential irrigation strategy. Table S6 shows the parameters of three irrigation techniques (Raes et al., 2017). We can adjust the simulated ET and Y according to the performance of the irrigation schedule."

**Table S6.** Parameters of three irrigation techniques.

| Irrigation technique | From day | Time criterion | Depth criterion | Water quality | Soil surface wetted |
| --- | --- | --- | --- | --- | --- |
| | | Depleted RAW | Back to FC | $E_{cw}$ | |
| | | (%) | (+/- mm) | (dS m$^{-1}$) | (%) |
| Furrow | 1 | 50 | 10 | 1.5 | 80 |
| Micro | 1 | 20 | 10 | 0 | 40 |
| Sprinkler | 1 | 50 | 10 | 1.5 | 100 |

5. The most common abbreviation in water footprint assessment literature for water footprint is WF not WFP. I strongly suggest to stick to WF.

**Response:** Thank you very much for your valuable advice. We will change the abbreviation of water footprint in the manuscript to WF entirely.

**Specific comments**

6. The abstract should mention what method (model) has been used to estimate WFPs.

**Response:** Thank you very much for your suggestion. The following will be added to the abstract according to your suggestion.

"AquaCrop model with the outputs of six GCMs in Coupled Model Intercomparison Project Phase 5 (CMIP5) as its input data was used to simulate the WF of maize and wheat."

7. "Wheat WFP will increase under RCP2.6 (by 12 % until the 2080s), while decrease by 12 % under RCP8.5 until the 2080s." Please add a brief explanation for this opposite trend under RCP8.5 in the abstract.

**Response:** Thank you very much for your comment. The reason is that the $CO_2$ concentration in 2080s under RCP8.5 is higher, which leads to a higher increase in wheat yield and decrease in wheat WF. We will add this to the abstract.

8. Please add in the abstract what benchmarks have been determined. You mention that "Furthermore, the spatial distributions of WFP benchmarks will not change as dramatically as those of WFP" but the WFP benchmarks themselves have not been mentioned earlier in the abstract.

**Response:** Thank you very much for your suggestion. The following information will be added to the abstract according to your suggestion.

"WF benchmarks of maize and wheat in the humid zone are 13–32 % higher than those in the arid zone. The differences in WF benchmarks among various irrigation

techniques are more significant in the arid zone, which can be as high as 57 percent, for WF benchmarks for the 20th production percentile of sprinkler-irrigated and micro-irrigated wheat in 2013. The changes in WF and its benchmarks will be similar in response to future climate change. Nevertheless, WF benchmarks will not change as dramatically as WF in the same area, especially the area with limited agricultural development."

9. "The present study demonstrated that … must be addressed and monitored". Stated too strongly. Did you really provide evidence that this must be done (in order to …)?

**Response:** We are very sorry for our unclear expression and use of incorrect word. We will modify L.25-L.28 in the abstract to the following content according to your suggestion.

"The present study demonstrated that the visible different responses to climate change in terms of crop water consumption, water use efficiency, and WF benchmarks under different irrigation techniques cannot be ignored."

10. A general overview of the methodological steps at the start of the section is missing. You now jump directly into "Determining the baseline year", but it is not yet clear that/why you need to determine that (and why you use the Aridity Index for that).

**Response:** We deeply appreciate your valuable comment. We will add the Section 2.1 Research set-up, which provides a general overview of the methodological steps, at the beginning of Section 2 Method and data. The content is as follows.

**"2.1 Research set-up**

We studied the spatiotemporal responses of blue and green WF and corresponding WF benchmarks for two crops (maize and wheat) to future climate change under two climate change scenarios (RCP2.6 and RCP8.5), four different planting modes (rain-fed and furrow-, micro-, and sprinkler-irrigated). Firstly, we need to determine the

baseline year. Secondly, we consider different planting modes to quantify WF and corresponding WF benchmarks of two crops in the baseline year and future year levels under two climate change scenarios. Finally, the spatiotemporal responses of crop WF and corresponding WF benchmarks to future climate change are analyzed."

[Figure]

**Figure 1.** Flow chart for the study.

Meanwhile, the following will be added to Section 2.2 (original 2.1) Determining the baseline year according to your suggestion.

"To ensure that the simulation results of future climate change scenarios are still reliable and meaningful, we need to determine the baseline year. Climate determines the annual variability of WF (Zhuo et al., 2014). The baseline year should be determined when there is a relative balance between aridity and moist. Thence, the aridity index (AI) was used here."

11. Why do you take the maximum root depth (Zx) and Harvest Index (HI) from Allan et al. (1998)? These parameters are also available for maize and wheat

in the default crop files that come with AquaCrop, like the rest of the
parameters that you take from Raes et al. (2017).

**Response:** We are very sorry for our unclear expression and use of incorrect word. The
reference manual of AquaCrop (Raes et al., 2017) does have default maximum root
depth (Zx) and Harvest Index (HI). Since AquaCrop model itself is developed by Food
and Agriculture Organization of the United Nations (FAO). In order to make the model
simulation more reliable, we reset the Zx according to the FAO-56 recommendation
(Allan et al., 1998). In addition, we further combined the literature research on maize
and wheat in China to reset the HI. In this way, we make the crop parameters more in
line with the actual situation in China.

12. Refrain from mentioning that in your study the AquaCrop model was coupled
    with GCMs. You did not couple these models. You used GCM outputs as input
    for AquaCrop. That is something different than coupling models.

**Response:** We are very sorry for our unclear expression and deeply appreciate your
valuable comment. We will modify L. 404 in Section 4 Conclusions to the following
content according to your suggestion.

"AquaCrop model with the outputs of six GCMs in CMIP5 as its input data was used
to simulate the WF of maize and wheat."

13. In the before last sentence of the conclusion you suddenly introduce other
    agricultural management practices that water-saving irrigation technology to
    reduce agricultural water use, such as mulching. The way it is phrased suggest
    that this is a conclusion from this study, which is not the case. Thus, you may
    want to rephrase this. Also, it is advised to add in the Introduction a description
    on the alternative options to reduce agricultural water use, after which you
    decide to focus this study on exploring the effects of water-saving irrigation
    technology only.

**Response:** Thank you very much for your valuable comment. We will delete the

relevant inappropriate expression in Section 4 Conclusions according to your suggestion. Meanwhile, we will add the consideration to Section 3.5 Discussion on the choice of two WF benchmarking methods and different agricultural management practices in combination with your General comment 1. The content is as follows.

[revised manuscript text omitted]

---

## Author Comment (AC2)

**Authors' responses to interactive comments on "Spatiotemporal responses in crop water footprint and benchmark under different irrigation techniques to climate change scenarios in China"**

Dear Referee #2,

We appreciate very much your valuable and helpful comments and suggestions concerning our manuscript. We have studied all the comments carefully and responded as followed.

Anonymous Referee #2

The paper is interesting for the wide area it involves and the relevance of the area for corn and wheat production, despite results and methods are not totally novel

**Response:** We deeply appreciate your recognition of this study and valuable comment. Our research adopts the widely used method (Mekonnen and Hoekstra, 2011; Zhuo et al., 2016a, 2016b, 2019; Wang et al., 2019; Mialyk et al., 2022) to calculate the water footprint per unit crop (WF, the abbreviation of water footprint was changed from WFP to WF at Referee #1's suggestion). It does have the potential to continue to innovate. And we will also improve the research method in the follow-up study.

Meanwhile, we want to highlight the innovations in our research. Compared with existing researches, the innovative aspects are embodied in two points. The present study firstly clarifies large-scale spatiotemporal responses of WF to future climate change scenarios under different irrigation techniques. Although Wang et al. (2019) considered different irrigation techniques when calculating the large-scale WF of wheat. But he focused on only one crop, wheat, and did not consider the impact of future climate change. In addition, our research is also the first to explore large-scale changes

in WF benchmarks under future climate change scenarios.

There are also some unique conclusions in our research. We find that micro irrigation and sprinkler irrigation result in the lowest increases in WF for maize and wheat, respectively. Hence, these water-saving irrigation practices effectively mitigate the negative impact of climate change. Moreover, we find that crop WF benchmarks will not change as dramatically as WF in the same area, especially the area with limited agricultural development, which also proves the stability of WF benchmarks. These conclusions also contribute to the improvement of the existing WF research field.

Specific comments:

It's not clear how irrigation techniques scenario are managed in the analysis. Are the actual techniques implemented when baseline year was determined? And what about future scenarios? Are techniques assumed considering the actual feasibility of the territory?

**Response:** We are very sorry for our unclear expression and deeply appreciate your valuable comment. We obtained the planted areas of each crop under each irrigation technique at provincial level from 2000 to 2014 from the China Statistical Yearbook (NBSC, 2021). In this way, the proportion of irrigated area under different irrigation techniques in the baseline year 2013 can be calculated. Then the irrigated and rain-fed areas of maize and wheat at a 5-arc minute grid resolution from MIRCA2000 dataset (Portmann et al., 2010) were divided into different parts under various irrigation techniques. Since we mainly focus on the impact of future climate change on WF and corresponding benchmarks, the change in land use is not considered. We assumed that the crop planted areas will not change in the future compared to baseline year 2013.

Furthermore, we realized that description of different irrigation techniques settings was missing in Section 2.3 (original 2.2, since we added the Section 2.1 Research set-up at Referee #1's suggestion) Water footprint per unit crop calculation. Therefore, the

following will be added at the end of Section 2.3. And the relevant table will be placed in the supplementary material.

"In the simulation, we considered different planting modes, namely rain-fed and three different irrigation techniques (furrow, micro, and sprinkler irrigation). The irrigation schedule of three irrigation techniques in model was Generation of Irrigation Schedule, namely the generation of an irrigation schedule by specifying a time and depth criterion for planning or evaluating a potential irrigation strategy. Table S6 shows the parameters of three irrigation techniques (Raes et al., 2017). We can adjust the simulated ET and Y according to the performance of the irrigation schedule."

**Table S6.** Parameters of three irrigation techniques.

| Irrigation technique | From day | Time criterion | Depth criterion | Water quality | Soil surface wetted |
|---|---|---|---|---|---|
| | | Depleted RAW | Back to FC | $E_{cw}$ | |
| | | (%) | (+/- mm) | (dS m$^{-1}$) | (%) |
| Furrow | 1 | 50 | 10 | 1.5 | 80 |
| Micro | 1 | 20 | 10 | 0 | 40 |
| Sprinkler | 1 | 50 | 10 | 1.5 | 100 |

L. 148 "using temperature inputs and the Penman-Monteith method". Penman-Monteith equation requires solar radiation, wind spped, relative humidity, and pressure in order to compute potential evapotranspiration (PET). Can you clarify how you computed PET?

**Response:** We deeply appreciate and agree with your valuable comment. We will apply the Penman-Monteith equation you mentioned to compute reference evapotranspiration ($ET_o$) if future climate data types are sufficient, including monthly maximum and minimum air temperature ($T_{max}$ and $T_{min}$), actual vapour pressure ($e_a$), net radiation ($R_n$) and wind speed measured at 2 m ($u_2$). However, due to the limited future climate data obtained from the Climate Change, Agriculture and Food Security (CCAFS) database (Navarro-Racines et al., 2020; CCAFS, 2015), only monthly air temperature and

precipitation data were available. Therefore, FAO Penman-Monteith method with missing data was used here to compute $ET_0$ (Allen et al., 1998). There are corresponding procedures to estimate missing humidity, radiation and wind speed data in this method.

1. Estimating missing humidity data

Where humidity data are lacking or are of questionable quality, an estimate of actual vapour pressure ($e_a$, kPa) can be obtained by assuming that dewpoint temperature ($T_{dew}$, °C) is near minimum air temperature ($T_{min}$, °C). This statement implicitly assumes that at sunrise, when the air temperature is close to $T_{min}$, that the air is nearly saturated with water vapour and the relative humidity is nearly 100%. If $T_{min}$ is used to represent $T_{dew}$ then:

$$e_a = e^0(T_{min}) = 0.611 exp\left[\frac{17.27 T_{min}}{T_{min}+237.3}\right] ,\tag{1}$$

2. Estimating missing radiation data

We can estimate $R_n$ by combining elevation, latitude and longitude data and air temperature data for each grid. The net radiation ($R_n$, MJ m$^{-2}$ day $^{-1}$) is the difference between the incoming net shortwave radiation ($R_{ns}$, MJ m$^{-2}$ day $^{-1}$) and the outgoing net longwave radiation ($R_{nl}$, MJ m$^{-2}$ day $^{-1}$):

$$R_n = R_{ns} - R_{nl} ,\tag{2}$$

where $R_{ns}$ and $R_{nl}$ are calculated by Equations (3) and (4), respectively.

$$R_{ns} = (1 - \alpha)R_s ,\tag{3}$$

$$R_{nl} = \sigma\left[\frac{T_{max,K}^4 + T_{min,K}^4}{2}\right]\left(0.34 - 0.14\sqrt{e_a}\right)\left[1.35\frac{R_s}{R_{so}} - 0.35\right] ,\tag{4}$$

where $\alpha$ is the albedo or canopy reflection coefficient, which is 0.23 for the hypothetical grass reference crop, $R_s$ (MJ m$^{-2}$ day $^{-1}$) is the incoming solar radiation

(Equation 5), $\sigma$ (4.903 10$^{-9}$ MJ K$^{-4}$ m$^{-2}$ day$^{-1}$) is Stefan-Boltzmann constant, $T_{max,K}$ and $T_{min,K}$ (K=°C+273.6) are maximum and minimum absolute temperature during the 24-hour period, respectively, and $R_{so}$ (MJ m$^{-2}$ day$^{-1}$) is the clear-sky radiation (Equation 6).

Rs is estimated using Hargreaves' radiation formula based on the difference between the maximum and minimum air temperature:

$$R_s = K_{RS}\sqrt{(T_{max} - T_{min})R_a} \ , \tag{5}$$

where $K_{RS}$ (°C$^{-0.5}$) is the adjustment coefficient from 0.16 to 0.19, which differs for 'interior' or 'coastal' regions. For 'interior' locations, where land mass dominates and air masses are not strongly influenced by a large water body, $K_{RS} \approx 0.16$. For 'coastal' locations, situated on or adjacent to the coast of a large land mass and where air masses are influenced by a nearby water body, $K_{RS} \approx 0.19$. And $R_a$ (MJ m$^{-2}$ day$^{-1}$) is the extraterrestrial radiation (Equation 7).

Rso is estimated according to the elevation for each grid:

$$R_{so} = (0.75 + 2 10^{-5}Z)R_a \ , \tag{6}$$

where $Z$ (m) is the elevation above sea level.

Ra is estimated by Equation 7:

$$R_a = \frac{24(60)}{\pi} G_{sc}d_r[\omega_s \sin(\varphi)\sin(\delta) + \cos(\varphi)\cos(\delta)\sin(\omega_s)] \ , \tag{7}$$

where $G_{sc}$ (0.0820 MJ m$^{-2}$ min$^{-1}$) is the solar constant, $d_r$ is the inverse relative distance Earth-Sun (Equation 8), $\omega_s$ (rad) is the sunset hour angle (Equation 10), $\varphi$ (rad) is the latitude, and $\delta$ (rad) is the solar decimation (Equation 9).

$$d_r = 1 + 0.033\cos\left(\frac{2\pi}{365}J\right) \ , \tag{8}$$

$$\delta = 0.409 sin \left( \frac{2\pi}{365} J - 1.39 \right) , \tag{9}$$

$$\omega_s = \arccos[-tan(\varphi)\tan(\delta)] , \tag{10}$$

where J is the number of the day in the year between 1 (1 January) and 365 or 366 (31 December).

3. Estimating missing wind speed data

Where no wind data are available within the region, a value of 2 m s$^{-1}$ can be used as a temporary estimate for $u_2$. This value is the average over 2000 weather stations around the globe.

Finally, if necessary, we will provide the code scripts related to ET$_0$ computing.

I am not an English native speaker but I think there's need to make the manuscript be checked for English grammar.

**Response:** We deeply appreciate your valuable advice and apologize for our improper use of English grammar. If we are lucky enough to get a valuable chance to revise, we will carefully examine the grammar in the manuscript and ask for English native speaker's opinion to modify.

**References**

Allen, R. G., Pereira, L. S., Raes, D., and Smith, M.: Crop evapotranspiration-Guidelines for computing crop water requirements-FAO Irrigation and drainage paper 56, 300, FAO, Rome, Italy, 1998.

CCAFS: CCAFS-Climate Statistically Downscaled Delta Method data, Climate Change, Agriculture and Food Security, available at: www.ccafs-climate.org, 2015.

Mekonnen, M. M. and Hoekstra, A. Y.: The green, blue and grey water footprint of crops and derived crop products, Hydrol. Earth Syst. Sci., 15, 1577–1600,

https://doi.org/10.5194/hess-15-1577-2011, 2011.

Mialyk, O., Schyns, J. F., Booij, M. J., and Hogeboom, R. J.: Historical simulation of maize water footprints with a new global gridded crop model ACEA, Hydrol. Earth Syst. Sci., 26, 923–940, https://doi.org/10.5194/hess-26-923-2022, 2022.

Navarro-Racines, C., Tarapues, J., Thornton, P., Jarvis, A., and Ramirez-Villegas, J.: High-resolution and bias-corrected CMIP5 projections for climate change impact assessments, Sci. Data, 7, 7, https://doi.org/10.1038/s41597-019-0343-8, 2020.

NBSC: National Data, China, National Bureau of Statistics, Beijing, China, available at: https://data.stats.gov.cn/, 2021.

Portmann, F. T., Siebert, S., and Döll, P.: MIRCA2000-Global monthly irrigated and rainfed crop areas around the year 2000: A new high-resolution data set for agricultural and hydrological modelling, Global Biogeochem. Cy., 24, https://doi.org/10.1029/2008gb003435, 2010.

Raes, D., Steduto, P., Hsiao, T. C., and Fereres, E.: Reference manual, Chapter 2, AquaCrop model, Version 6.0, Food and Agriculture Organization of the United Nations, Rome, Italy, 2017.

Wang, W., Zhuo, L., Li, M., Liu, Y., and Wu, P.: The effect of development in water-saving irrigation techniques on spatial-temporal variations in crop water footprint and benchmarking, J. Hydrol., 577, https://doi.org/10.1016/j.jhydrol.2019.123916, 2019.

Zhuo, L., Liu, Y., Yang, H., Hoekstra, A. Y., Liu, W., Cao, X., Wang, M., and Wu, P.: Water for maize for pigs for pork: An analysis of inter-provincial trade in China, Water Res., 166, https://doi.org/10.1016/j.watres.2019.115074, 2019.

Zhuo, L., Mekonnen, M. M., and Hoekstra, A. Y.: The effect of inter-annual variability of consumption, production, trade and climate on crop-related green and blue water footprints and inter-regional virtual water trade: A study for China (1978-

2008), Water Res., 94, 73–85, https://doi.org/10.1016/j.watres.2016.02.037, 2016a.

Zhuo, L., Mekonnen, M. M., Hoekstra, A. Y., and Wada, Y.: Inter- and intra-annual variation of water footprint of crops and blue water scarcity in the Yellow River basin (1961–2009), Adv. Water Resour., 87, 29–41, https://doi.org/10.1016/j.advwatres.2015.11.002, 2016b.

---

## Author Response (AR1)

**Spatiotemporal responses in crop water footprint and benchmark under different irrigation techniques to climate change scenarios in China**

Zhiwei Yue, Xiangxiang Ji, La Zhuo, Wei Wang, Zhibin Li, Pute Wu

**Authors' responses to Referees' comments**

We appreciate very much for the opportunity to revise the study and the valuable comments and suggestions by the Editor and two Referees. We have carefully addressed all the comments and provided our detailed responses below them, responding point by point. The revised parts are colored in RED in the revised manuscript.

**Editor**

Comments to the author:

The manuscript has been revised by two highly qualified Reviewers. Both Reviewers consider the topic of the paper of potential interest for HESS readers and provide very detailed comments that I think will be extremely useful for the authors. Reviewer 2 also noticed that the methodologies employed in the current manuscript do not appear to be entirely novel.

If the authors decide to re-submit the manuscript to HESS, in addition to the reviewers' comments, I encourage the authors to:

- Highlight the key novel aspects of their work (in terms of monitoring or data analysis, process concepts, technology, and/or theoretical aspects).

- Revise the use of English language and grammar

**Response:** We are very grateful for the valuable opportunity to revise and improve the manuscript according to constructive comments and suggestions by Editor and two highly qualified Referees. We carefully address each of the comments and respond point by point.

-Specifically, in order to highlight the novel aspects of the current study, we revise the relative text in the end Introduction (Lines 101–104). "Compared to existing literatures, the innovations of the current research are embodied in two points. The present study clarifies large-scale spatiotemporal responses of WF to future climate change scenarios under different irrigation techniques for the first time. This analysis is also the first to explore large-scale changes in WF benchmarks under future climate change scenarios." Combined with revisions according to corresponding comment by Referee#1 to clarify the pros and cons of the methodology to setting WF benchmarks, we believe that the innovative aspects and unique results are clearer in the revision.

-Thank you very much for the suggestion on language editing. We asked for a third part English editing service to improve the language in the latest submission. They also helped to check the original version of the manuscript. We believe that the language issues are addressed in the revision. Please kindly find attached the editing certificate in the end of the document.

Thank you again for the chance of revision.

**General comments**

1. One could divide WFP benchmarking techniques into two methods. Method 1 compares the WFP of different producers (or grid cells) within the same region, ranks them and sets a benchmark based on some percentile. Method 2 compares the WFP at each location under different management practices and sets a benchmark based on best practices (those resulting in the smallest WF). Method 2 is for example applied in the studies by Chukalla et al. (https://doi.org/10.5194/hess-21-3507-2017 and https://doi.org/10.5194/hess-19-4877-2015). The drawback of method 1 is that no matter what spatial scope one takes in grouping producers, within that scope there will still be variability from place to place (section 4.3.2.1 in https://doi.org/10.1016/B978-0-12-822112-9.00006-0). Rainfall, for example, shows strong spatial variability over short distances, such that a few producers in a larger area simply had more favourable local circumstances. Therefore, one can always question the comparability of producers that operate in different locations and the WFPs they achieve. Method 2 overcomes this drawback. In this manuscript method 1 is applied, although different irrigation practices are simulated. What are your reasons for determining the benchmarks based on method 2? Why don't you determine the benchmarks (also) based on method 1? You seem to have the data/simulations for that.

**Response:** We deeply appreciate your valuable comment. Sure, we acknowledge that three are two methods of establishing WF benchmarks (Hoekstra, 2013). The reason why we chose Method 1 instead of Method 2 is that we mainly explore the responses of large-scale WFs for two grain crops to future climate changes under specific irrigation technique, that is, each irrigation technique has its corresponding WF benchmarks. If Method 2 was selected, what we concerned about was the responses of WF to future climate change under the optimal tillage plus irrigation techniques which would result in the smallest WF. It is inconsistent with the current research objectives. This is also relevant to your General comment 2. Method 2 has the higher requirements on the setting and simulation of different agricultural management practices. However, we focus on only one agricultural management practice here, i.e., irrigation. If we chose Method 2, the calibration of different agricultural management practices would be the key. The existing data cannot meet these requirements for such a large-scale study. Therefore, we choose Method 1 to determine WF benchmarks. And we also realized that the application of Method 1 does have some limitations. We add the consideration to Section 3.5 Discussion (Lines 415–433) on the choice of two WF benchmarking methods and different agricultural management practices in combination with your Specific comment 13. The content is as follows.

"Three are two methods of establishing WF benchmarks (Hoekstra, 2013). Method 1 is based on yield accumulation statistical analysis. Due to the variability of WFs found across regions and among producers within a region, for each crop, we can select the WF of 20 % or 25 % of the producers with the highest water productivity as the WF benchmark (Mekonnen and Hoekstra, 2014). Method 2 is based on the available optimal technique analysis. We can compare the WFs at each location under different agricultural management practices and take the WF associated with optimal practice, which results in the smallest WF, as the WF benchmark (Chukalla et al., 2015). Both methods establish WF benchmarks based on the maximum reasonable water consumption in each step of the product's supply chain (Hoekstra, 2014). Method 1 is suitable for large-scale application. The differences in environmental conditions (such as climate) and development conditions should be considered comprehensively (Mekonnen and Hoekstra, 2014; Zhuo et al., 2016a). The drawback of Method 1 is that no matter what spatial scope one takes in grouping producers, within that scope there will still be variability from place to place even if the differences in regional environmental and development conditions are taken into

account (Schyns et al., 2022). Method 2 is suitable for smaller scale and overcomes this drawback of Method 1 to some extent. The Method 2's drawback is that it has the higher requirements on the setting and simulation of different agricultural management practices. We mainly want to explore the response of large-scale WF to future climate change under specific irrigation technique, that is, each irrigation technique has its corresponding WF benchmarks. And only one agricultural management practice, that is irrigation, is considered here. Therefore, we choose Method 1. A combination of methods should be established. If conditions permit, we strongly recommend that Method 1 and Method 2 are combined to establish small-scale WF benchmarks. Different agricultural management practices, such as irrigation, mulching techniques and so on, can be combined to further determine WF benchmarks."

2. AquaCrop provides crop parameters sets for maize and wheat which are to some degree calibrated for the conditions of recent history. How do you make sure the model produces reliable results for ET and Y under climate change scenarios?

**Response:** We are very sorry for our unclear expression and use of incorrect words. To some extent, we guaranteed how evapotranspiration (ET) and yield (Y) will develop in the climate change scenarios at the current production level. First, we chose the year 2013 as the baseline year, when the drought level came closest to the 15-year national average drought level over 2000–2014. Second, the simulated Y per grid for each crop in 2013 was calibrated via scaling model simulation outputs to accord with the crop yield statistics data at provincial level (NBSC, 2021), which was consistent with the widely used calibration method (Mekonnen and Hoekstra, 2011; Zhuo et al., 2016b, 2016c, 2019; Wang et al., 2019; Mialyk et al., 2022). For sure, the calibrated Y corresponded to the simulated ET. The crop parameters in the model represent the existing agricultural production level. Climate was the only variable for future scenario simulations. We add the above information in the Method (Lines 156–161).

3. Micro-irrigation results in the smallest WFP and largest Y (Figure 3). Yet how feasible (and profitable) is micro irrigation in maize and wheat production in practice? Is it commonly applied for these crops in some parts of the world? Or is micro-irrigation mostly used for cash-crops only? Some elaboration on this in the manuscript is needed to justify the research setup and to put the results into perspective.

**Response:** We appreciate your valuable suggestion. Due to the high cost and technical requirements, micro irrigation applications are limited and are mostly applied to cash crops. However, there is a serious shortage of water resources in some croplands in China. And the spatial and temporal distribution of water and soil resources is uneven. Developing water-saving irrigation has become an important way to alleviate the prominent contradiction between water resources utilization and grain production in China. According to NBSC (2021), the area of water-saving irrigation projects in China in 2019 was 37 million ha, including 7 million ha for micro irrigation. Therefore, micro irrigation does apply to food crops in China despite the limited irrigated area. For instance, in Xinjiang province, the area of micro irrigated maize and wheat was 0.033 million ha in 2009 (CIDDC, 2022), of which the wheat area dominated at up to 0.031 million ha (Wang et al., 2011). Meanwhile, some scholars are conducting research on micro irrigated maize (Bai and Gao, 2021; Guo et al., 2021) and wheat (Li et al., 2021; Zain et al., 2021) in China, especially in the North. We add the above information in the end Introduction (Lines 93–99).

4. What assumption do you take in terms of irrigation strategy/scheduling? This needs to be added to the methods. And how does this affect your results? This is important to address in the discussion, preferably with some quantitative substantiation. The more irrigation events you have, the more effect you will see from moving to a more efficient irrigation application technology (from furrow to drip). So I suppose your outcomes in terms of WFP for different irrigation technologies are quite sensitive to the assumption for the irrigation trigger (x% of

soil moisture depletion?) and amount (back to field capacity?).

**Response:** We are very sorry for our unclear expression and deeply appreciate your valuable comment. We realized that description of different irrigation techniques settings was missing in Section 2.3 (original 2.2) Water footprint per unit crop calculation. Therefore, the following information is added at the end of Section 2.3 (Lines 162–166). The relevant Table S6 has been placed in the supplementary material.

"In the simulation, different planting modes, namely rain-fed and three different irrigation techniques (furrow, micro, and sprinkler irrigation), were considered. The irrigation schedule of three irrigation techniques in the model was the Generation of Irrigation Schedule, namely the generation of an irrigation schedule by specifying a time and depth criterion for planning or evaluating a potential irrigation strategy. Table S6 shows the parameters of three irrigation techniques (Raes et al., 2017). We can adjust the simulated ET and Y according to the performance of the irrigation schedule."

**Table S6.** Parameters of three irrigation techniques.

| Irrigation technique | From day | Time criterion | Depth criterion | Water quality | Soil surface wetted |
| --- | --- | --- | --- | --- | --- |
| | | Depleted RAW | Back to FC | Ecw | |
| | | (%) | (+/- mm) | (dS m$^{-1}$) | (%) |
| Furrow | 1 | 50 | 10 | 1.5 | 80 |
| Micro | 1 | 20 | 10 | 0 | 40 |
| Sprinkler | 1 | 50 | 10 | 1.5 | 100 |

5. The most common abbreviation in water footprint assessment literature for water footprint is WF not WFP. I strongly suggest to stick to WF.

**Response:** Thank you very much for your valuable advice. We correct the abbreviation of water footprint in the manuscript to WF entirely.

**Specific comments**

6. The abstract should mention what method (model) has been used to estimate WFPs.

**Response:** Thank you very much for your suggestion. The following sentence is added to the abstract (Lines 19–21) according to your suggestion.

"The AquaCrop model with the outputs of six global climate models in Coupled Model Intercomparison Project Phase 5 (CMIP5) as its input data was used to simulate the WF of maize and wheat."

7. "Wheat WFP will increase under RCP2.6 (by 12 % until the 2080s), while decrease by 12 % under RCP8.5 until the 2080s." Please add a brief explanation for this opposite trend under RCP8.5 in the abstract.

**Response:** Thank you very much for your comment. The reason is that the $CO_2$ concentration in 2080s under RCP8.5 is higher, which leads to a higher increase in wheat yield and decrease in wheat WF. We add this reason to the abstract (Lines 24–25).

8. Please add in the abstract what benchmarks have been determined. You mention that "Furthermore, the spatial distributions of WFP benchmarks will not change as dramatically as those of WFP" but the WFP benchmarks themselves have not been mentioned earlier in the abstract.

**Response:** Thank you very much for your suggestion. The following information is added to the abstract (Lines 27–31) according to your suggestion.

"The WF benchmarks of maize and wheat in the humid zone are 13–32 % higher than those in the arid zone. The differences in WF benchmarks among various irrigation techniques are more significant in the arid zone, which can be as high as 57%, for 20th percentile WF benchmarks of sprinkler-irrigated and micro-irrigated wheat. Nevertheless, WF benchmarks will not respond to climate changes as dramatically as the WF in the same area, especially in the area with limited agricultural development."

9. "The present study demonstrated that … must be addressed and monitored". Stated too strongly. Did you really provide evidence that this must be done (in order to …)?

**Response:** We are very sorry for our unclear expression and use of incorrect word. We have modified this sentence in the abstract (Lines 31–33) to the following information according to your suggestion.

"The present study demonstrated that the visible different responses to climate change in terms of crop water consumption, water use efficiency, and WF benchmarks under different irrigation techniques cannot be ignored."

10. A general overview of the methodological steps at the start of the section is missing. You now jump directly into "Determining the baseline year", but it is not yet clear that/why you need to determine that (and why you use the Aridity Index for that).

**Response:** We deeply appreciate your valuable comment. We have added the Section 2.1 Research set-up, which provided a general overview of the methodological steps, at the beginning of Section 2 Method and data (Lines 109–119). The content is as follows.

**"2.1 Research set-up**

We studied the spatiotemporal responses of blue and green WF and corresponding WF benchmarks for two crops (maize and wheat) to future climate change under two climate change scenarios (RCP2.6 and RCP8.5) using four different planting modes (rain-fed and furrow-, micro-, and sprinkler-irrigated). First, we determined the baseline year. Second, we considered different planting modes to quantify WF and corresponding WF benchmarks of two crops in the baseline year and future year levels under two climate change scenarios. Finally, the spatiotemporal responses of crop WF and corresponding WF benchmarks to future climate change were analysed (Fig. 1)."

[Figure]

**Figure 1.** Flow chart for the study.

Meanwhile, the following information is added to Section 2.2 (original 2.1) Determining the baseline year (Lines 121–123) according to your suggestion.

"To ensure that the simulation results of future climate change scenarios are still reliable and meaningful, the baseline year was determined. Climate determines the annual variability of WF (Zhuo et al., 2014), and the baseline year should be determined when there is a relative balance between aridity and moisture. Hence, the aridity index (AI) was used here."

> 11. Why do you take the maximum root depth (Zx) and Harvest Index (HI) from Allan et al. (1998)? These parameters are also available for maize and wheat in the default crop files that come with AquaCrop, like the rest of the parameters that you take from Raes et al. (2017).

**Response:** We are very sorry for our unclear expression and use of incorrect word. The reference manual of AquaCrop (Raes et al., 2017) does have default maximum root depth (Zx) and Harvest Index (HI). Since AquaCrop model itself is developed by Food and Agriculture Organization of the United Nations (FAO). To make the model simulation more reliable, we reset the Zx according to the FAO-56 recommendation (Allan et al., 1998). In addition, we further combined the literature research on maize and wheat in China to reset the $HI_0$ (Zhuo et al., 2016c). In this way, we make the crop parameters more in line with the actual situation in China. We modify the corresponding sentences in Section 2.5 (original 2.4) Data sources (Lines 186–188) to the following information according to your suggestion.

"To make the model simulation more in line with the actual situation in China, we reset the maximum root depth (Zx) according to the FAO-56 recommendation (Allan et al., 1998). In addition, we further combined the literature research on maize and wheat in China to reset the $HI_0$ (Zhuo et al., 2016c)."

> 12. Refrain from mentioning that in your study the AquaCrop model was coupled with GCMs. You did not couple these models. You used GCM outputs as input for AquaCrop. That is

something different than coupling models.

**Response:** We are very sorry for our unclear expression and deeply appreciate your valuable comment. We delete the relevant inappropriate expression in the last paragraph of Section 3.5 Discussion and modify the related sentence in Section 4 Conclusions (Lines 461–462).

"The AquaCrop model with the outputs of six GCMs in CMIP5 as its input data was used to simulate the WF of maize and wheat."

13. In the before last sentence of the conclusion you suddenly introduce other agricultural management practices that water-saving irrigation technology to reduce agricultural water use, such as mulching. The way it is phrased suggest that this is a conclusion from this study, which is not the case. Thus, you may want to rephrase this. Also, it is advised to add in the Introduction a description on the alternative options to reduce agricultural water use, after which you decide to focus this study on exploring the effects of water-saving irrigation technology only.

**Response:** Thank you very much for your valuable comment. We delete the relevant inappropriate expression in Conclusions according to your suggestion. Meanwhile, we add the consideration to Section 3.5 Discussion (Lines 415–433) on the choice of two WF benchmarking methods and different agricultural management practices in combination with your General comment 1. In the Introduction, we add the explanation on the reasons and feasibility of exploring the effects of different water-saving irrigation technologies (Lines 93–99), which is also referring to your valuable General comment 3.

Thank you again for your efforts and time on improving our study substantially.

There are also unique and new conclusions. We find that micro irrigation and sprinkler irrigation result in the lowest increases in WF for maize and wheat, respectively, to future possible climate changes. Hence, these water-saving irrigation practices will effectively mitigate the negative impacts of climate changes. Moreover, we find that crop WF benchmarks will not change as dramatically as WF in the same area, especially in the area with limited agricultural development, which also proves the stability of WF benchmarks. We believe these conclusions also contribute to the improvement of the existing WF research field.

**Specific comments:**

*It's not clear how irrigation techniques scenario are managed in the analysis. Are the actual techniques implemented when baseline year was determined? And what about future scenarios? Are techniques assumed considering the actual feasibility of the territory?*

**Response:** We are very sorry for our unclear expression and deeply appreciate your valuable comment. We obtained the planted areas of each crop under each irrigation technique at provincial level from 2000 to 2014 from the China Statistical Yearbook (NBSC, 2021). In this way, the proportion of irrigated area under different irrigation techniques in the baseline year 2013 can be calculated. Then the irrigated and rain-fed areas of maize and wheat at a 5-arc minute grid resolution from MIRCA2000 dataset (Portmann et al., 2010) were divided into different parts under various irrigation techniques. Since we mainly focus on the impact of future climate change on WF and corresponding benchmarks, the change in land use is not considered. We assumed that the crop planted areas will not change in the future compared to baseline year 2013.

Furthermore, we realized that description of different irrigation techniques settings was missing in Section 2.3 (original 2.2, since we added the Section 2.1 Research set-up at Referee #1's suggestion) Water footprint per unit crop calculation. Therefore, the following information is added at the end of

Section 2.3 (Lines 162–166).

"In the simulation, different planting modes, namely rain-fed and three different irrigation techniques (furrow, micro, and sprinkler irrigation), were considered. The irrigation schedule of three irrigation techniques in the model was the Generation of Irrigation Schedule, namely the generation of an irrigation schedule by specifying a time and depth criterion for planning or evaluating a potential irrigation strategy. Table S6 shows the parameters of three irrigation techniques (Raes et al., 2017). We can adjust the simulated ET and Y according to the performance of the irrigation schedule."

**Table S6.** Parameters of three irrigation techniques.

| Irrigation technique | From day | Time criterion | Depth criterion | Water quality | Soil surface wetted |
|---|---|---|---|---|---|
| | | Depleted RAW | Back to FC | Ecw | |
| | | (%) | (+/- mm) | (dS m$^{-1}$) | (%) |
| Furrow | 1 | 50 | 10 | 1.5 | 80 |
| Micro | 1 | 20 | 10 | 0 | 40 |
| Sprinkler | 1 | 50 | 10 | 1.5 | 100 |

L. 148 "using temperature inputs and the Penman-Monteith method". Penman-Monteith equation requires solar radiation, wind spped, relative humidity, and pressure in order to compute potential evapotranspiration (PET). Can you clarify how you computed PET?

**Response:** We deeply appreciate and agree with your valuable comment. We will apply the Penman-Monteith equation you mentioned to compute reference evapotranspiration ($ET_0$) if future climate data types are sufficient, including monthly maximum and minimum air temperature ($T_{max}$ and $T_{min}$), actual vapour pressure ($e_a$), net radiation ($R_n$) and wind speed measured at 2 m ($u_2$). However, due to the limited future climate data obtained from the Climate Change, Agriculture and Food Security (CCAFS) database (Navarro-Racines et al., 2020; CCAFS, 2015), only monthly air temperature and precipitation data were available. Therefore, FAO Penman-Monteith method with missing data was used here to compute $ET_0$ (Allen et al., 1998). We modify the relevant sentence in Section 2.5 (original 2.4) Data sources (Lines 183–185) to be clearer.

"As the CCAFS database has no $ET_0$ data, we calculated $ET_0$ for each climate scenario using temperature inputs via the FAO Penman-Monteith method with missing data as described by Allen et al. (1998)."

There are corresponding procedures to estimate missing humidity, radiation and wind speed data in this method.

1. Estimating missing humidity data

Where humidity data are lacking or are of questionable quality, an estimate of actual vapour pressure ($e_a$, kPa) can be obtained by assuming that dewpoint temperature ($T_{dew}$, °C) is near minimum air temperature ($T_{min}$, °C). This statement implicitly assumes that at sunrise, when the air temperature is close to $T_{min}$, that the air is nearly saturated with water vapour and the relative humidity is nearly 100%. If $T_{min}$ is used to represent $T_{dew}$ then:

$$e_a = e^0(T_{min}) = 0.611 \exp\left[\frac{17.27 T_{min}}{T_{min}+237.3}\right] \ , \tag{1}$$

2. Estimating missing radiation data

We can estimate $R_n$ by combining elevation, latitude and longitude data and air temperature data for each grid. The net radiation ($R_n$, MJ m$^{-2}$ day$^{-1}$) is the difference between the incoming net shortwave radiation ($R_{ns}$, MJ m$^{-2}$ day$^{-1}$) and the outgoing net longwave radiation ($R_{nl}$, MJ m$^{-2}$ day$^{-1}$):

$$R_n = R_{ns} - R_{nl} \ , \tag{2}$$

where $R_{ns}$ and $R_{nl}$ are calculated by Equations (3) and (4), respectively.

$$R_{ns} = (1-\alpha)R_s \ , \tag{3}$$

$$R_{nl} = \sigma\left[\frac{T_{max,\,K}^4 + T_{min,\,K}^4}{2}\right](0.34 - 0.14\sqrt{e_a})\left[1.35\frac{R_s}{R_{so}} - 0.35\right] \ , \tag{4}$$

where $\alpha$ is the albedo or canopy reflection coefficient, which is 0.23 for the hypothetical grass reference crop, $R_s$ (MJ m$^{-2}$ day$^{-1}$) is the incoming solar radiation (Equation 5), $\sigma$ (4.903 10$^{-9}$ MJ K$^{-4}$ m$^{-2}$ day$^{-1}$) is Stefan-Boltzmann constant, $T_{max,K}$ and $T_{min,K}$ (K=°C+273.6) are maximum and minimum absolute temperature during the 24-hour period, respectively, and $R_{so}$ (MJ m$^{-2}$ day$^{-1}$) is the clear-sky radiation (Equation 6).

Rs is estimated using Hargreaves' radiation formula based on the difference between the maximum and minimum air temperature:

$$R_s = K_{RS}\sqrt{(T_{max}-T_{min})R_a} \ , \tag{5}$$

where $K_{RS}$ (°C$^{-0.5}$) is the adjustment coefficient from 0.16 to 0.19, which differs for 'interior' or 'coastal' regions. For 'interior' locations, where land mass dominates and air masses are not strongly influenced by a large water body, $K_{RS} \approx 0.16$. For 'coastal' locations, situated on or adjacent to the coast of a large land mass and where air masses are influenced by a nearby water body, $K_{RS} \approx 0.19$. And $R_a$ (MJ m$^{-2}$ day$^{-1}$) is the extraterrestrial radiation (Equation 7).

Rso is estimated according to the elevation for each grid:

$$R_{so} = (0.75 + 210^{-5}Z)R_a \ , \tag{6}$$

where $Z$ (m) is the elevation above sea level.

Ra is estimated by Equation 7:

$$R_a = \frac{24(60)}{\pi} G_{sc} d_r [\omega_s \sin(\varphi) \sin(\delta) + \cos(\varphi)\cos(\delta)\sin(\omega_s)] \ , \tag{7}$$

where $G_{sc}$ (0.0820 MJ m$^{-2}$ min$^{-1}$) is the solar constant, $d_r$ is the inverse relative distance Earth-Sun (Equation 8), $\omega_s$ (rad) is the sunset hour angle (Equation 10), $\varphi$ (rad) is the latitude, and $\delta$ (rad) is the solar decimation (Equation 9).

$$d_r = 1 + 0.033\cos\left(\frac{2\pi}{365}J\right) \ , \tag{8}$$

$$\delta = 0.409\sin\left(\frac{2\pi}{365}J - 1.39\right) \ , \tag{9}$$

$$\omega_s = \arccos[-\tan(\varphi)\tan(\delta)] \ , \tag{10}$$

where $J$ is the number of the day in the year between 1 (1 January) and 365 or 366 (31 December).

3. Estimating missing wind speed data

Where no wind data are available within the region, a value of $2 \text{ m s}^{-1}$ can be used as a temporary estimate for $u_2$. This value is the average over 2000 weather stations around the globe.

I am not an English native speaker but I think there's need to make the manuscript be checked for English grammar.

**Response:** We deeply appreciate your valuable advice and apologize for our improper use of English grammar. We asked for native English speaker through a language editing service to enhance the text. Please kindly refer to our response to the Editor's comments.

Thank you again for your efforts and time on improving our study.

**References**

[revised manuscript text omitted]

---

## Author Response (AR2)

**Spatiotemporal responses in crop water footprint and benchmark under different irrigation techniques to climate change scenarios in China**

Zhiwei Yue, Xiangxiang Ji, La Zhuo, Wei Wang, Zhibin Li, Pute Wu

**Authors' responses to Referees' comments**

We are very grateful to Editor Professor Monica Riva and Referees for the opportunity of minor revision and the valuable comments and suggestions. We have carefully addressed all the comments and provided our detailed responses below them, responding point by point. The revised parts are colored in RED in the revised manuscript.

**Editor comments:**

Comments to the author:

The Authors have seriously taken in consideration the main comments of the reviews and provided a stronger revised manuscript. Some additional (minor) changes are still requested in order to better convey the main message of the manuscript. In this spirit, I encourage the authors to further review their work. Then I will be in the position to finalize the assessment of the manuscript.

**Response:** We thank Editor very much for handling our submission and constructive suggestions on improving the study. We carefully address each of the following comments and revise the manuscript accordingly.

**Anonymous referee #1:**

Dear authors,

You did a good job in responding to the reviewer comments and revising the manuscript. I have a few more follow-up questions based on your responses to my earlier comments, namely:

**Response:** Thank you very much for the positive words!

a) About your response to comment #2: What you call a "calibration method" isn't really a calibration, but rather a post-simulation correction (scaling) of Y. The method itself is fine and indeed widely applied, but I suggest to not call it a calibration, which suggests that you changed certain (sensitive) parameters in the model with the goal of improving some objective function value. Furthermore, I suspect that you make the implicit assumption that the crop parameters (e.g. the WP* parameter in AquaCrop) for the current/historic situation are still applicable in climate change scenarios. I think you can assume that, but

**Response:** We deeply appreciate your valuable comment. We correct the word "calibration" into "scaling" accordingly (Line 159). The crop parameters which were assumed as being consistent in the climate change scenarios are clarified in the text (Line 162).

"With the consistent scaling factors for the Y simulation and crop parameters including the crop calendar, WP*, $HI_0$, and the maximum root depth which represent the existing agricultural production level, climate was the only variable for future scenario simulations."

b) About your response to comment #4: It is good that you added this information and Table S6. I think Table S6 should definitely be part of the main manuscript (not the supplement), since it contains important information to interpret your results. I also have a couple of questions about Table S6's contents:

**Response:** As suggested, we move Table S6 from the supplementary material to the manuscript as current Table 2 (Line 175).

i) What depth criterion did you use? Back to field capacity (FC) when the time criterion is reached? Or add 10 mm when the time criterion is reached?

**Response:** The depth criterion we used was the "Back to field capacity (+/- mm)". The value of Back to field capacity used here was 10 mm, which means the extra 10 mm of water on top of the amount of irrigation water required to bring the root zone back to field capacity (Raes et al., 2017). We clarify the text (Line 173) and the explicit information is now available in Table 2.

ii) You should explain your choices for the specific values that you put in for the time criterion, the water quality and the surface area wetted. Did you base this on literature? For example, why is irrigation applied earlier (lower trigger value), and the water quality better, in the case of micro irrigation?

**Response:** The parameters values of three irrigation techniques were selected according to the reference manual of AquaCrop model (Raes et al., 2017). The time criterion we used was Allowable depletion (%), namely the percentage of the Readily Available soil Water (RAW) that can be depleted before irrigation water has to be applied. The Water quality was expressed by the Electrical conductivity (dS m$^{-1}$) of the irrigation water. The Soil surface wetted (%), an indicative value for the fraction of soil surface wetted, can be used to select irrigation techniques. We add the explanation in the text (Lines 167-171). Table 2 shows the values of each of above parameters for different modeling scenarios. The Allowable depletion and Electrical conductivity of micro irrigation were lower. Due to the lower Soil surface wetted, micro irrigation usually starts earlier, which results in the lower Allowable depletion. The equipment of micro irrigation has the higher requirement for water quality, which results in the lower Electrical conductivity.

**Table 2.** Parameters of three irrigation techniques.

| Irrigation technique | From day | Time criterion | Depth criterion | Water quality | Soil surface wetted |
| --- | --- | --- | --- | --- | --- |
| | | Allowable depletion | Back to field capacity | Electrical conductivity | |
| | | (%) | (+/- mm) | (dS m$^{-1}$) | (%) |
| Furrow | 1 | 50 | 10 | 1.5 | 80 |
| Micro | 1 | 20 | 10 | 0 | 40 |
| Sprinkler | 1 | 50 | 10 | 1.5 | 100 |

c) About your response to comment #8: Sorry, I wasn't clear in the formulation of my comment. I wanted to see the absolute benchmark values that you have estimated, e.g. "xxx m³/tonne for sprinkler-irrigated and xxx m³/tonne for micro-irrigated wheat". Can you add that information in the abstract?

**Response:** Accordingly, we add pointed absolute benchmark values in the abstract. The relative text is as below (Lines 28-31).

"The WF benchmarks of maize and wheat in the humid zone (~overall average at 680 m$^3$ t$^{-1}$ for maize and 873 m$^3$ t$^{-1}$ for wheat at 20$^{th}$ percentile) are 13–32 % higher than those in the arid zone (~overall average at 601 m$^3$ t$^{-1}$ for maize and 753 m$^3$ t$^{-1}$ for wheat). The differences in WF benchmarks among various irrigation techniques are more significant in the arid zone, which can be as high as 57%, for 20th percentile WF benchmarks of 1020 m$^3$ t$^{-1}$ for sprinkler-irrigated wheat and 648 m$^3$ t$^{-1}$ for micro-irrigated wheat."

d) About your response to comment #10: Figure 1 is a nice addition. I only find the term "Planting modes" strange. That sounds to me like different ways to plant the seeds. Would the term "Growing modes" or "Irrigation modes" (with rain-fed being "no irrigation (rain-fed)") be better? Furthermore, you state that: "To ensure that the simulation results of future climate change scenarios are still reliable and meaningful, the baseline year was determined." I don't understand what you intend to say here. Can you rephrase? Setting a baseline year (or average of years) is needed for a comparison between future and current conditions, bur setting a baseline doesn't affect the scenarios (so doesn't make them more reliable or meaningful).

**Response:** We are very sorry for our unclear expression and use of incorrect word. We correct the "planting modes" into "growing modes" in both Figure 1 and text. In addition, we realize that the sentence "To ensure that the simulation results of future climate change scenarios are still reliable and meaningful, the baseline year was determined." in Section 2.2 Determining the baseline year is inaccurate and modify it to the following sentence (Line 124).

"Determining the baseline year is needed for a comparison between future and current conditions."

[Figure]

**Figure 1.** Flow chart for the study.

e) About your response to comment #11: I totally understand now why you updated HIo, but I have my doubts whether this FAO-56 recommendation (from 1998) for Zx is better than the Zx given in the calibrated AquaCrop crop parameter files for wheat and maize. Is there a large difference between those values (FAO-56 vs. AquaCrop)? And if so, why would you think this rather old reference value is better?

**Response:** We deeply appreciate your valuable comment. Following Table R1 shows the comparison of the maximum root depth ($Z_x$) for maize and wheat between FAO-56 recommendation (Allan et al., 1998), AquaCrop manual (Raes et al., 2017) and our study. The FAO-56 recommendation gives a reference range for the $Z_x$, which enables to distinguish the differences between irrigated and rainfed fields. While the AquaCrop manual files only give the upper criteria for certain crops. Furthermore, the differences between winter wheat and spring wheat are not shown in the model default parameter values. Therefore, we prefer to use the FAO-56 recommendations. We add the brief explanation in the text (Lines 197-198).

**Table R1.** The comparison of the maximum root depth ($Z_x$) between FAO-56 recommendation, AquaCrop mannual and the current study.

| Crop | $Z_x$ (m) | | | |
| --- | --- | --- | --- | --- |
| | FAO-56 Report | AquaCrop Mannual | Current study | |
| | | | Irrigated | Rain-fed |
| Maize | 1.0 ~ 1.7 | Up to 2.80 | 1 | 1.7 |
| Spring Wheat | 1.0 ~ 1.5 | Up to 2.40 | 1 | 1.5 |
| Winter Wheat | 1.5 ~ 1.8 | Up to 2.40 | 1.5 | 1.8 |

f) Finally, I have one more comment in response to your updated novelty statement: You say that "this

analysis is also the first to explore large-scale changes in WF benchmarks under future climate change scenarios." I think you can nuance that statement. There are several studies that assessed the WF under climate change scenarios. See for example the recent study by Karandish et al. (https://doi.org/10.1029/2021EF002095) and some of the references therein.

**Response:** We deeply appreciate your valuable comment. Karandish et al. (2022) also assess the effectiveness of three adaptation strategies, off-season cultivation, early planting, and water footprint (WF) benchmarking, on Iran's blue water savings in future. While the differences between different irrigation techniques were not considered. We clarify the text accordingly (Lines 102–106).

"Compared to existing literatures on evaluation of WFs of crop production under climate change scenarios (e.g., Karandish et al., 2022), the innovations of the current research are embodied in two points. The present study clarifies large-scale spatiotemporal responses of WF to future climate change scenarios under different irrigation techniques for the first time. This analysis is also the first to explore the large-scale future changes in WF benchmarks under different irrigation techniques."

I hope these comments are helpful in making some minor revisions to this manuscript.

**Response:** Thank you again for your efforts and valuable comments!

**References**

Karandish, F., Nouri, H., and Schyns, J. F.: Agricultural adaptation to reconcile food security and water sustainability under climate change: the case of cereals in Iran, Earths Future, https://doi.org/10.1029/2021EF002095, 2022.

**References**

Allen, R. G., Pereira, L. S., Raes, D., and Smith, M.: Crop evapotranspiration-Guidelines for computing crop water requirements-FAO Irrigation and drainage paper 56, 300, FAO, Rome, Italy, 1998.

Karandish, F., Nouri, H., and Schyns, J. F.: Agricultural adaptation to reconcile food security and water sustainability under climate change: the case of cereals in Iran, Earth's Future, 10, e2021EF002095, https://doi.org/10.1029/2021EF002095, 2022.

Raes, D., Steduto, P., Hsiao, T. C., and Fereres, E.: Reference manual, Chapter 2, AquaCrop model, Version 6.0, Food and Agriculture Organization of the United Nations, Rome, Italy, 2017.